# Lattice Convolutional Networks for Learning Ground States of Quantum Many-Body Systems

## Abstract

Deep learning methods have been shown to be effective in representing ground-state wave functions of quantum many-body systems. Existing methods use convolutional neural networks (CNNs) for square lattices due to their image-like structures. For non-square lattices, the existing method uses graph neural networks (GNNs) in which structure information is not precisely captured, thereby requiring additional hand-crafted sublattice encoding. In this work, we propose lattice convolutions in which a set of proposed operations are used to convert non-square lattices into grid-like augmented lattices on which regular convolution can be applied. Based on the proposed lattice convolutions, we design lattice convolutional networks (LCN) that use self-gating and attention mechanisms. Experimental results show that our method achieves performance on par or better than the GNN method on spin $1/2$ $J_1$-$J_2$ Heisenberg model over the square, honeycomb, triangular, and kagome lattices while without using hand-crafted encoding.

## 1 Introduction

The study of quantum many-body problems is of fundamental interest in physics. It is crucial for the theoretical modeling and simulation of complex quantum systems, materials, and molecules (Carleo et al., 2019). For instance, graphene, arguably the most famous 2D material, is made of carbon atoms on a honeycomb lattice. Solving quantum many-body problems remains to be very challenging because of the exponential growth of Hilbert space dimensions with the number of particles in quantum systems. Only approximation solutions are available in most cases. Tensor network (White, 1992; Schollwöck, 2011; Orús, 2014; Biamonte & Bergholm, 2017) is one of the popular techniques to model quantum many-body systems but suffers entanglement problems (Choo et al., 2018). Variational Monte Carlo (VMC) (McMillan, 1965) is a more general methodology to obtain quantum many-body wave functions by optimizing a compact parameterized variational ansatz with data sampled from itself. But how to design variational ansatz with high expressivity to represent real quantum states is still an open problem.

Recently traditional machine learning models, such as restricted Boltzmann machine (RBM) (Smolensky, 1986), have been used as variational ansatz (Carleo & Troyer, 2017; Nomura et al., 2017; Choo et al., 2018; Kaubruegger et al., 2018; Choo et al., 2020; Nomura, 2021; Chen et al., 2022). Following this direction, some studies explore deep Boltzmann machines (Gao & Duan, 2017; Carleo et al., 2018; Pastori et al., 2019) and fully-connected neural networks to represent quantum states (Saito & Kato, 2018; Cai & Liu, 2018; Saito, 2017; 2018; Saito & Kato, 2018). Most recent studies also use CNN as variational ansatz for square lattice systems (Liang et al., 2018; Choo et al., 2019; Zheng et al., 2021; Liang et al., 2021; Roth & MacDonald, 2021). Moreover, GNN has been applied to non-square lattices and random graph systems (Yang et al., 2020; Kochkov et al., 2021).

In this work, we explore the potential of using CNN as a variational anstaz for non-square lattice quantum spin systems. We propose lattice convolutions that use a set of proposed operations to convert non-square lattices into grid-like augmented lattices on which any existing CNN architectures can be applied. Based on the proposed lattice convolution, we design highly expressive lattice convolutional networks (LCN) by leveraging self-gating and attention mechanisms. Experimental results show that our method achieves performance on

par or better than the GNN method over the square, honeycomb, triangular, and kagome lattice quantum systems on spin $1/2$ $J_1$-$J_2$ Heisenberg model, a prototypical quantum many-body model of magnetic materials that captures the exchange interaction between spins.

**Novelty and Significance.** Our work proposes the first pure deep learning approach that does not require any prior knowledge of quantum physics to solve quantum many-body problems on different types of lattice systems. Our approach overcomes the shortcomings of previous neural quantum state methods, which not only require extensive prior knowledge but are also designed for a specific lattice or even a specific regime. However, our method can be seamlessly applied to different lattices and can still achieve competitive or even better performance than existing methods without introducing prior knowledge. As a result, our method possesses great generalizability in practice. This makes our approach of great value in the study of quantum many-body problems.

**Relations with Prior Work.** GNN (Kochkov et al., 2021) was proposed as the first and generic method that can be applied to various lattice shapes. To this end, it is natural that LCN uses the same experiment setting as GNN. While GNN uses different hand-crafted sublattice encoding techniques for different lattice structures, LCN only needs to augment different lattices in a simple and principled way without any prior knowledge. This significantly enhances the generalization capability of LCN in practice. Roth & MacDonald (2021) proposed a general framework called Group-CNN. However, it can only be easily applied to square and triangular lattices. Moreover, it still needs to consider specific symmetry groups for different lattice systems as prior knowledge. Choo et al. (2019) applies CNN on the square lattice, but it needs to use specific quantum physics knowledge such as point group symmetry and the Marshall sign rule, which is the known sign structure of the ground state. However, the Marshal sign rule only works for bipartite graphs (such as square lattices) and non-frustrated regimes. In geometric deep learning, there were also some studies applying CNN to irregular structures such as triangle meshes (Hanocka et al., 2019; Hu et al., 2022) to perform 3D shape analysis. These works utilize the unique properties of triangular meshes and design specialized convolutional neural networks that operate on the meshes by leveraging their intrinsic connections. However, in our task, we want to apply CNN on different kinds of lattices instead of triangle meshes so these methods aren't directly applicable to quantum many-body problems on lattice systems.

## 2 Background and Related Work

In quantum mechanics, a quantum state is represented as a vector in Hilbert space. This vector is a linear combination of observable system configurations $\{c_i\}$, known as a computational basis. In the context of spin $1/2$ systems, each spin can be measured in two states, spin-up or spin-down, which are represented by $\uparrow$ and $\downarrow$, respectively. All the combinations of spins form a basis. Given $N$ spins, there are in total $2^N$ configurations in the computational basis. Specifically, a state can be written as

$$|\psi\rangle = \sum_i^{2^N} \psi(c_i)|c_i\rangle, \tag{1}$$

where $|c_i\rangle$ represents an array of spin configurations of $N$ spins, e.g., $\uparrow\uparrow\downarrow\cdots\downarrow$, and $\psi(c_i)$ is the wave function value or probability amplitude of the configuration $c_i$, which is in general a complex number. The summation is over all possible $2^N$ spin configurations. The squared norm $|\psi(c_i)|^2$ corresponds to the probability of system collapsing to configuration $c_i$ when being measured, and $\sum_i^{2^N} |\psi(c_i)|^2 = 1$ due to normalization.

### 2.1 Ground States

The ground state of a quantum system is its lowest-energy state. Usually, many physical properties can be determined by the ground state. Particle interactions within a given quantum many-body system are determined by a Hamiltonian, which is a Hermitian matrix $H$ in the Hilbert space. System energy and its corresponding quantum state are governed by the time-independent Schrödinger equation:

$$H|\psi\rangle = E|\psi\rangle, \tag{2}$$

which is an eigenvalue equation. The eigenenergy $E$ is the eigenvalue of $H$ and $|\psi\rangle$ is the corresponding eigenvector. In principle, those can be obtained by eigenvalue decomposition given $H$. The lowest eigenvalue is called the ground state energy, and its associated eigenvector is called the ground state. The ground state and the ground state energy determine the property of the quantum system at zero temperature.

## 2.2 Variational Principle in Quantum Mechanics

Given a system of size $N$, the dimension of the Hamiltonian matrix is $2^N \times 2^N$. Since the dimension of the matrix grows exponentially with system size, it is intractable to use eigenvalue decomposition directly, even for relatively small systems. The state-of-the-art algorithm using the Lanczos method, which explores the sparseness of $H$, can obtain the ground state energy and the ground state for $N$ up to $\sim 48$. For larger systems, a common approach is to use the variational principle to approximately solve the Schrödinger equation. According to the variational principle, the energy of any given quantum state is greater than or equal to the ground state energy. So we can optimize parameterized wave functions to make the energy as low as possible. Specifically, we can approximate the ground state of a Hamiltonian $H$ by minimizing the variational term $E$ shown below:

$$E = \frac{\langle\psi|H|\psi\rangle}{\langle\psi|\psi\rangle} \geq E_0, \tag{3}$$

where $E$ is the expectation value of the energy of a variational quantum state $|\psi\rangle$ for a given Hamiltonian $H$, and $E_0$ is the true ground state energy. The state $|\psi\rangle$ takes the form of Eq. equation 1. Given $H$, the expectation value $E$ is determined by the wavefunction $\psi(c_i)$ that can be any parameterized function. The goal is to find the optimal function $\psi(c_i)$ that minimizes $E$. The success of the variational method relies on the expressivity of the parameterized function. Therefore it is natural to explore neural networks as variational ansatz of the wavefunction.

## 2.3 Related Work

Variational quantum many-body states with wave functions given by neural networks are called neural-network quantum states, initially studied by Carleo & Troyer (2017), where they use RBM to represent many-body wave function. Subsequent studies (Choo et al., 2018; Saito, 2018; Cai & Liu, 2018) apply fully-connected neural networks as variational ansatz, which has been shown to be more effective than RBM methods. But these methods do not explicitly consider the structure information when applied to two-dimensional systems (Cai & Liu, 2018). Motivated by the spatial symmetry of periodic quantum systems and successful practices of convolutional neural networks (CNNs) in computer vision (Krizhevsky et al., 2012), Liang et al. (2018), Choo et al. (2019) and Szabó & Castelnovo (2020) use CNNs to represent quantum states for square lattices. Albergo et al. (2021) and Boyda et al. (2021) combine flow-based sampling with CNN for lattice field theories. Favoni et al. (2022) proposes lattice gauge equivariant CNN for machine learning problems on lattice gauge theory. CNN is able to effectively represent highly entangled quantum systems (Levine et al., 2019) than RBM-based representations (Deng et al., 2017), which benefits from its information reuse. However, CNN cannot be naturally used on non-grid like systems. Recently graph neural networks (GNNs) have also been applied to represent wave functions (Yang et al., 2020; Kochkov et al., 2021). GNNs can work with arbitrary geometric lattices or even random graph systems, but structure information is not precisely captured. So additional hand-crafted sublattice encoding is needed to augment system configurations in order to respect underlying quantum symmetry (Kochkov et al., 2021).

Wave functions are usually complex-valued, therefore, it is necessary to predict both amplitudes and phases. Choo et al. (2019) use complex-valued weights and biases to predict amplitudes and phases simultaneously and design a generalized activation function. Whereas amplitudes and phases are predicted separately using real-valued networks in Szabó & Castelnovo (2020); Kochkov et al. (2021).

## 3 Lattice Convolutional Networks

While GNN can be naturally applied to non-square lattices, isotropic weights prevent it from capturing rich structure information. Therefore auxiliary hand-crafted structure encoding is needed to augment the original

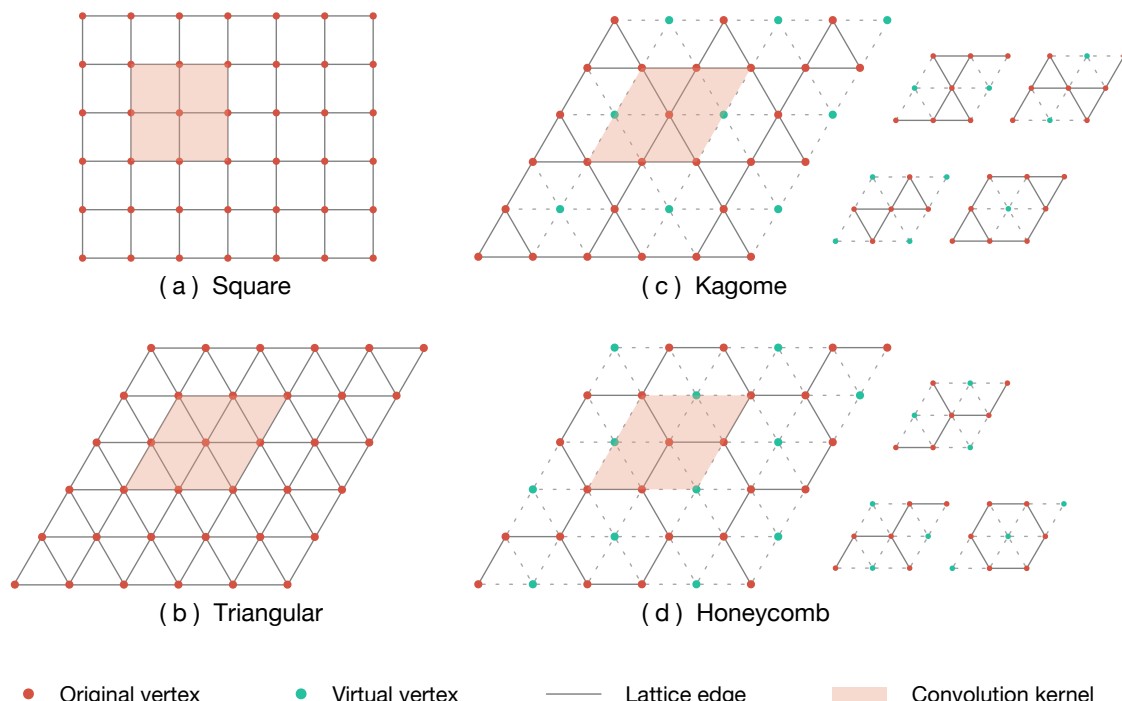

Figure 1: Structures of four different lattices studied in this work. Shadow areas denote the regular convolution kernels. Red dots represent original vertices in lattices, and green dots represent padded virtual vertices. In (c) and (d), dashed lines denote the nearest neighbors of virtual vertices. On the right are different substructures captured by convolution. Note that shear directions won't affect the result due to lattice symmetry. Even though the lattices shown are finite but they are periodically arranged in whole space, which is realized by applying the periodic boundary condition.

spin configuration input on lattices. We argue that CNN is more suitable to model wave functions for lattice systems, which feature repetitive local patterns.

In this section, we introduce LCN, a novel lattice convolutional network that has a strong capability to model wave functions for non-square lattice systems without using any extra structure encoding.

### 3.1 Motivation and Overview

In this work, we focus on designing CNNs on four types of lattice, including square, triangular, honeycomb, and kagome, which are the four most common lattice structures that describe two-dimensional materials. The lattice structures are shown in Figure 1. CNNs are known to be efficient feature extractors on regular structures. Inside the network, each convolution layer applies anisotropic filters on local regions shared across the entire input. Through training, each filter learns to be sensitive to different local patterns. By stacking multiple convolution layers, the global structure information can be precisely captured.

It is straightforward to apply CNNs on a square lattice due to its image-like structure. Triangular lattices can be viewed as sheared square lattices where every unit cell of the square lattice undergoes the same affine transformation. Therefore we can shear the convolution kernel accordingly to match the shape of the transformed unit cells. However, for lattices that cannot be converted into grids, such as honeycomb and kagome, the key is to define the shape of the convolution kernel and optimize the weight sharing across convolution sites. In the proposed LCN, we solve these challenges in a principled way by converting non-square lattices into grid-like augmented lattices through a set of operations such that regular convolution kernels can be applied.

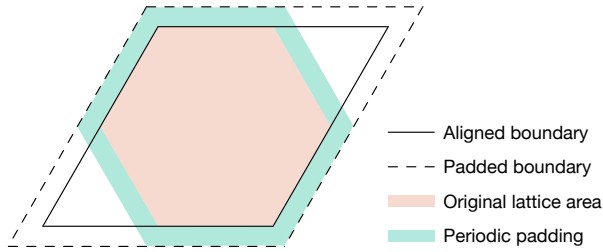

Figure 2: Boundary alignment and periodic padding.

## 3.2 Augmented Lattices

As mentioned in Section 3.1, triangular lattices can be seen as sheared square lattices, which implies that the local structure is the same everywhere on the lattice, hence regular square kernels can be naturally applied. By contrast, honeycomb and kagome lattices have multiple local structures, making it difficult to share kernel weights across different structures. Moreover, different local structures are arranged in a staggered manner, which impedes information reuse among the same local structures.

Critically, we make the key observation that honeycomb and kagome lattices can be converted from triangular lattices by removing some vertices and edges. Conversely, honeycomb and kagome lattices can be viewed as triangular lattices by augmenting virtual vertices back on the original lattices. As shown in Figure 1 (c) and (d), virtual vertices (green dots) are inserted in the center of each hexagon sub-structure. Through this augmenting operation, we can apply identical kernels everywhere on the lattices regardless of original local structures but still can capture different structure information.

The advantages of this operation are two folds. First, during convolution, the virtual vertices participate in the convolution in the same way as the original vertices. i.e., values of virtual vertices are also updated. By doing so, the virtual lattices can gather and distribute the information from the original vertices, which can help increase the receptive field and boost information exchange. Second, by adding virtual vertices, we can overlap the same convolution kernel in order to enable information reuse, which is crucial to capture long-range spin correlations (Liang et al., 2018). To some extent, augmenting with virtual vertices could enhance the wave function representation ability.

## 3.3 Lattice Convolutions

After augmentation, the lattice becomes either square or triangular, both of which are grid-structured. However, due to the characteristics of the input, additional processing steps are required in order to apply regular convolutions on the augmented lattice.

**Boundary Alignment (BA).** Lattices with finite vertices often have irregular boundaries. However, in current deep learning libraries, convolutions on image-like data typically require the input grid to have an equal number of elements on each row. Therefore, we zero-pad the augmented lattices into parallelograms. In our experiments, square lattices already have regular boundaries, so this step is omitted. The aligned boundary is drawn with the solid line in Figure 2.

**Periodic Padding (PP).** For images, boundaries are commonly zero-padded before convolutions to preserve the size of feature maps. Finite quantum systems often consider periodic boundary conditions so that the lattice can be repeated to fill the entire space. To preserve this important structure information, after padding the aligned boundaries with zero, we replace the padding values around the original lattice area with the values given by the periodic boundary condition. We can optionally do the periodic padding for the virtual vertices as well. In Figure 2, the padded boundary is drawn with the dashed line, the original lattice area is marked with a pink shadow, and the periodic padding is marked with a green shadow.

**Mask.** Finally, after each convolution, to clean up the artifacts introduced in the two previous steps, we reset all vertices used for the boundary alignment and the periodic padding to zero. We do not reset the

virtual vertices to allow information to pass through them, i.e., we only reset the vertices outside the original lattice area. We also conduct an ablation study of mask operation in Appendix H.1

To summarize, the proposed lattice convolution applied on an input lattice $U$ is defined as:

$$\text{LatticeConv}(U; W) = \mathbf{Mask}(W * \mathbf{PP}(\mathbf{BA}(\mathbf{Aug}(U)))), \tag{4}$$

where $W$ is the convolution weight matrix; $\mathbf{PP}$, $\mathbf{BA}$ and $\mathbf{Mask}$ stand for the above mentioned three processing steps and $\mathbf{Aug}$ stands for the augmentation step defined in Section 3.2; The symbol $*$ denotes the regular convolution which is defined as:

$$(W * U')_{i,j} = \sum_{m=-s}^{s} \sum_{n=-s'}^{s'} W_{mn} U'_{i-m,j-n}, \tag{5}$$

where $U' \in \mathbb{R}^{h \times w \times d}$ denotes a $h \times w$ feature map with $d$ input channels. $W$ has shape $(2s+1) \times (2s'+1) \times d' \times d$ where $d'$ is the number of output channels and $(2s+1) \times (2s'+1)$ is the size of the receptive field.

### 3.4 Instantiations

**Square.** As shown in Figure 1 (a), for square lattices, the $3 \times 3$ kernel receptive field contains all four nearest neighbors and four second nearest neighbors (or called next nearest neighbor in some references), which is defined by the euclidean distance, instead of by connectivity.

**Triangular.** For triangular lattices, the receptive field of $3 \times 3$ convolution kernel includes six nearest neighbors and two second nearest neighbors of center vertices, as shown in Figure 1 (b). This structure pattern is the same at all positions across the lattice.

**Honeycomb.** For honeycomb lattices, the $3 \times 3$ convolution kernel centered on the original vertices captures all three nearest neighbors and two second nearest neighbors. As shown in the Figure 1 (d), depending on the local structure, the mapping between the kernel weights and the original vertices can have two different situations. The kernel centered on virtual vertices captures six original vertices around it. For the honeycomb lattice, we also apply the periodic padding for the virtual vertices.

**Kagome.** For kagome lattices, as shown in Figure 1 (c), the $3 \times 3$ kernel can capture all nearest neighbors and some next nearest neighbors. The virtual vertices can receive information from eight original vertices around them. There are three local structures centered on original vertices. And we only apply the periodic padding for the virtual vertices on kagome lattices of size 36.

### 3.5 CNN versus GNN

The convolution applies different weights for neighbors with different relative positions, which we argue is critical to capture the structure information. This is the opposite with graph neural networks, where the same weights are used for all one-hop neighbors on a graph. For example, we can define a graph on the square lattice by defining edges as the spatially nearest neighbors. For quantum many-body systems, the spatially second nearest neighbors play an important role in defining the energy. But the two-hop neighbors on the graph will also include the third nearest neighbors. To be able to capture the structural information with GNNs, Kochkov et al. (2021) proposes to augment the vertices with sublattice encoding to explicitly provide structure information at the input. Our experiments show that such encoding is indeed critical for GNNs. On the contrary, our lattice convolution accurately learns the ground state without any additional input. To some extent, the proposed lattice convolution can learn the structure encoding automatically in the kernel space.

## 4 Network Architecture and Training

After constructing the grid-like input and defining the convolution operation, any existing convolution neural network architecture can be applied. We aim to design a variational ansatz to have powerful expressivity and

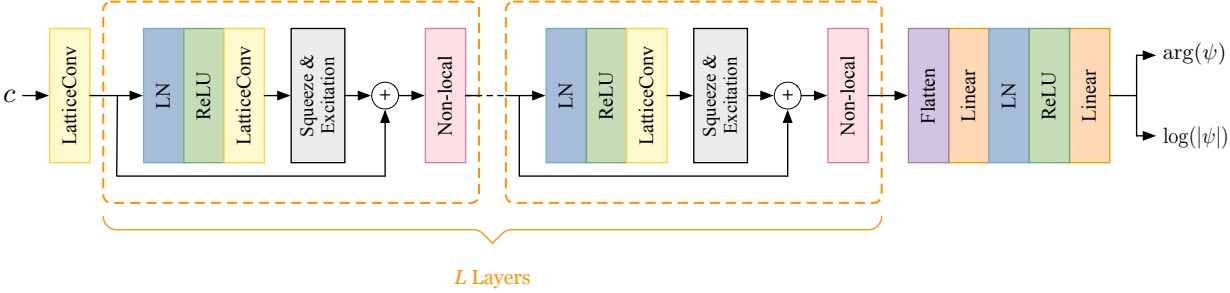

Figure 3: Lattice Convolutional Networks architecture. The network takes as input a specific spin configuration $c$ on the lattice and outputs the amplitude $\log(|\psi(c)|)$ and phase $\arg(\psi(c))$ of the complex wavefunction value $\psi(c)$.

capture spatial long-range spin correlations. To this end, our model is developed based on recently advanced deep learning modules and attention mechanisms. The details of the main components are described below.

**Squeeze-and-Excitation Block.** Squeeze-and-Excitation (SE) block (Hu et al., 2018) can improve the quality of representations produced by a network by explicitly modeling the channel-wise interdependencies. It utilizes squeeze operation to aggregate channel-wise global spatial information and excitation operation to capture channel-wise dependencies through self-gating recalibration. Details of the formulation can be found in Appendix F.

**Non-Local Block.** We find it useful to incorporate spin-spin global interaction other than only locality interaction defined by lattice edges. Non-local operation (Wang et al., 2018) is designed to capture long-range dependencies. Specifically, non-local operations make features at one position attend to all other position's features. Details of the formulation can be found in Appendix F.

**SE-Non-Local Layer.** Based on the SE block and non-local block, we propose SE-Non-Local Layer by further introducing skip connection and pre-activation techniques. Therefore, the resulting SE-Non-Local Layer consists of the following components: Normalization → Activation → LatticeConv → SE Block → Addition → Non-local Block. We apply LayerNorm (LN) (Ba et al., 2016) as the normalization and ReLU as the activation function.

In our model, we predict amplitudes and phases separately with the real-valued network, following Kochkov et al. (2021). We first use lattice convolution to transform the original spin configuration input into embedding space and then stack multiple SE-Non-Local layers to obtain the final latent representation. At last, we flatten the feature maps into a vector to keep all information on each lattice vertex and use MLP to obtain the log amplitude and argument of the wave function value. The overall architecture of our variational ansatz is shown in Figure 3.

**Training.** As discussed in Section 2.2, we use the Variational Monte Carlo approach to optimize the variational ansatz by minimizing the statistic expectation of system energy. The equation is shown below:

$$
\begin{aligned}
E = \frac{\langle\psi|\mathbf{H}|\psi\rangle}{\langle\psi|\psi\rangle} &= \frac{\sum_{i,j}\langle\psi|c_i\rangle\langle c_i|\mathbf{H}|c_j\rangle\langle c_j|\psi\rangle}{\langle\psi|\psi\rangle}, \\
&= \frac{\sum_i |\psi(c_i)|^2 \left(\sum_j H_{ij}\frac{\psi(c_j)}{\psi(c_i)}\right)}{\sum_i |\psi(c_i)|^2}, \\
&= E_{c_i \sim D}\left[\sum_j H_{ij}\frac{\psi(c_j)}{\psi(c_i)}\right],
\end{aligned}
\tag{6}
$$

where the energy expectation is computed over the probability distribution $D(c_i) = \frac{|\psi(c_i)|^2}{\sum_i |\psi(c_i)|^2}$, and $\sum_j H_{ij} \frac{\psi(c_j)}{\psi(c_i)}$ is refereed to as local energy. Specifically, at each optimization step, we first use Markov-Chain Monte Carlo (MCMC) method to sample system configurations $c_i$ from target distribution, which is given by the neural network or called the variational wave function. Then we can stochastically estimate the gradient $\nabla_w \langle E \rangle$ (Kochkov et al., 2021) to update network parameters $\mathbf{w}$. Meanwhile, we can evaluate the energy expectation in every optimization step by taking the average of all the local energy associated with each sampled configuration. The matrix multiplication between $H$ and $|\psi\rangle$ can be performed efficiently because of the sparseness of the Hamiltonian, which is determined by the lattice topology and the specific quantum system. Given a quantum system of $N$ spin-1/2, the dimension of $H$ is $2^N$, however, the typical number of nonzero values in each row is only of order $N$. The full training algorithm is shown below:

---

**Algorithm 1** Training Algorithm of Lattice Convolutional Networks

---
 1: **Input:** Lattice structure $\mathcal{L}$, number of spin sites $N$, Lattice convolution network $\psi_\theta$ with trainable parameter $\theta$, learning rate $\alpha$, Markov Chain batch size $B$, initial annealing step $s$, measure step $m$
 2: Set different random seeds for each Markov chain
 3: Randomly initialize spin configurations $\mathcal{C} \in \{+1, -1\}^{B \times N}$ with equal number of $+1$ and $-1$
 4: $\hat{\mathcal{C}} \leftarrow$ **Metropolis–Hastings**($\psi_\theta$, $\mathcal{C}$, s)
 5: **repeat**
 6: $\quad E_{total} \leftarrow 0$
 7: $\quad$ **for** $i = 1$ **to** $B$ **do**
 8: $\quad\quad E_{total} \leftarrow E_{total} + \sum_j H_{ij} \frac{\psi(c_j)}{\psi(c_i)}$
 9: $\quad$ **end for**
10: $\quad \hat{E}_0 \leftarrow \frac{1}{B} E_{total}$
11: $\quad \theta \leftarrow \theta - \alpha \nabla \hat{E}_0$
12: $\quad \hat{\mathcal{C}} \leftarrow$ **Metropolis–Hastings**($\psi_\theta$, $\hat{\mathcal{C}}$, m)
13: **until** $\hat{E}_0$ is converged

---

## 5 Experiments

In this section, we evaluate the proposed LCN on learning ground states of the spin-1/2 $J_1$-$J_2$ Heisenberg model, where the input vertices represent quantum spins. We show that our model can accurately approximate the ground state energies and achieve on par or better results with GNN models.

**Lattice.** Following Kochkov et al. (2021), four kinds of lattice are used in our experiments, including square, honeycomb, triangular, and kagome. Periodic boundary conditions are used so that the neighborhood patterns of the boundary vertices are the same as the internal vertices. All the lattice geometries we use in the experiments can be found in Kochkov et al. (2021, Appendix A.5.a).

**$J_1$-$J_2$ Quantum Heisenberg model.** The $J_1$-$J_2$ quantum Heisenberg model is the prototypical model for studying the magnetic properties of quantum materials. Its Hamiltonian matrix is given by:

$$H = \sum_{\langle i,j \rangle} \mathbf{S}_i \cdot \mathbf{S}_j + J_2 \sum_{\langle\langle i,j \rangle\rangle} \mathbf{S}_i \cdot \mathbf{S}_j, \tag{7}$$

where $\mathbf{S}_i = (S_i^x, S_i^y, S_i^z)$ are the spin-1/2 operators of the $i$-th vertex. The spin operator $S_i^\alpha$ is an Hermitian matrix of dimension $2^N$, defined as $S_i^\alpha = I^{\otimes j-1} \otimes \sigma^\alpha \otimes I^{\otimes N-j}$, where $\otimes$ stands for Kronecker product, $I$ is the two-by-two identity matrix and $\sigma^\alpha$ is the two-by-two Pauli matrix for $\alpha = x, y, z$. The term $\mathbf{S}_i \cdot \mathbf{S}_j$ describes the antiferromagnetic exchange between the spin on site $i$ and the spin on site $j$. $\langle \cdot, \cdot \rangle$ denotes the nearest neighbors, and $\langle\langle \cdot, \cdot \rangle\rangle$ denotes the second nearest neighbors, both in terms of Euclidean distances; $J_2$ controls the interaction strength between the next nearest neighbors. The interaction strength between the nearest neighboring spins is set to 1 as the unit.

**Setup.** We test our models on four kinds of lattice with various sizes and $J_2$ values. We compare the proposed LCN with GNN (Kochkov et al., 2021) as well as reference energies. We use the same references as

Table 1: Estimated energy per site of the learned wave function (with error bars in parenthesis). Lower is better. Four kinds of lattices with various sizes are used for comparison. The $J_2$ value controls the next nearest neighboring interaction on the lattice, resulting in differences in the ground states. The best result in each row is denoted in bold (if two are the same, we don't bold-face any results for clarity). We use $*$ to denote the results measured from plots in their reference papers. For the reference ground state energies, we also annotate the employed methods: † for exact diagonalization, $\infty$ for infinite-size estimates, ‡ for RMB+PP, § for QMC methods, and ¶ for DMRG methods.

| Lattice | Size | $J_2$ | GNN | LCN (ours) | Reference Energy |
|---|---|---|---|---|---|
| Square | 36 | 0 | -0.6788* | -0.6788(1) | -0.678872† |
| | | 0.5 | -0.5022(4) | -0.5022(2) | -0.503810† |
| | 100 | 0 | -0.6708(0) | -0.6708(1) | 0.671549(4)§ |
| | | 0.5 | -0.4955(4) | **-0.4957(4)** | -0.497629(1)‡ |
| Honeycomb | 32 | 0 | -0.551* | **-0.5516(1)** | -0.5517†* |
| | | 0.2 | -0.4563(6) | -0.4563(1) | -0.4567†* |
| | 98 | 0 | - | **-0.5421(4)** | -0.5448∞* |
| | | 0.2 | -0.4528(2) | **-0.4538(4)** | -0.4527∞* |
| Triangular | 36 | 0 | - | **-0.5601(4)** | -0.5603734† |
| | | 0.08 | -0.5221* | **-0.5273(4)** | -0.5286†* |
| | | 0.125 | -0.512(2) | **-0.5126(6)** | -0.515564† |
| | 108 | 0 | **-0.5508(8)** | -0.5475(4) | -0.551∞ |
| | | 0.125 | -0.500(9) | **-0.5110(4)** | -0.5126¶ |
| Kagome | 36 | 0 | -0.434(1) | **-0.4367(4)** | -0.43837653† |
| | | -0.02 | -0.4339* | **-0.4384(5)** | -0.4399†* |
| | 108 | 0 | **-0.4302(1)** | -0.4276(3) | -0.4386¶ |

in Kochkov et al. (2021). We also choose specific J2 values under which the ground state is much harder to learn for the GNN model. The **energy per site** of the learned wave function is used as an evaluation metric where the total energy of the system is divided by the number of spins (vertices). A lower energy per site indicates a more accurate approximation of the ground state. More implementation details can be found in Appendix B. Considering we are proposing a generic CNN network that fits multiple kinds of lattice without prior knowledge, we don't include other customized physics-incorporated networks for square lattice as baselines. But for completeness, we conduct experiments on full range J2 of square lattice to compare with other state-of-the-art models in Appendix I.

**Results.** The experimental results are summarized in Table 1. For small systems ($N = 32, 36$), the reference energies are the exact ground state energies computed by direct diagonalizing the Hamiltonian matrix (Schulz et al., 1996; Albuquerque et al., 2011; Iqbal et al., 2016; Changlani et al., 2018). For large systems ($N = 98, 108$), the exact diagonalization is computationally infeasible. Reference energies are calculated with quantum Monte Carlo (QMC) (Sandvik, 1997) and RBM+PP (pair-product states) (Nomura & Imada, 2021) for square with $J_2 = 0.0$ and $J_2 = 0.5$, respectively, and density-matrix renormalization group (DMRG) (Iqbal et al., 2016; Yan et al., 2011) for triangular and kagome. In some cases, the estimates for infinite-size lattice are used as reference energies, extrapolated from exact diagonalizations for honeycomb or from DMRG for triangular. These reference energy of large systems can be seen as a very tight upper bound of true ground state energy.

Results demonstrate that the wave functions learned by the proposed LCN consistently gives energies close to the reference ground state energies on both small and large systems. For small systems, LCN gives similar ground state energies with GNN on square and honeycomb lattices while achieving better ground state energies on triangular and kagome lattices. For large systems at $J_2 = 0$, LCN achieves similar ground state

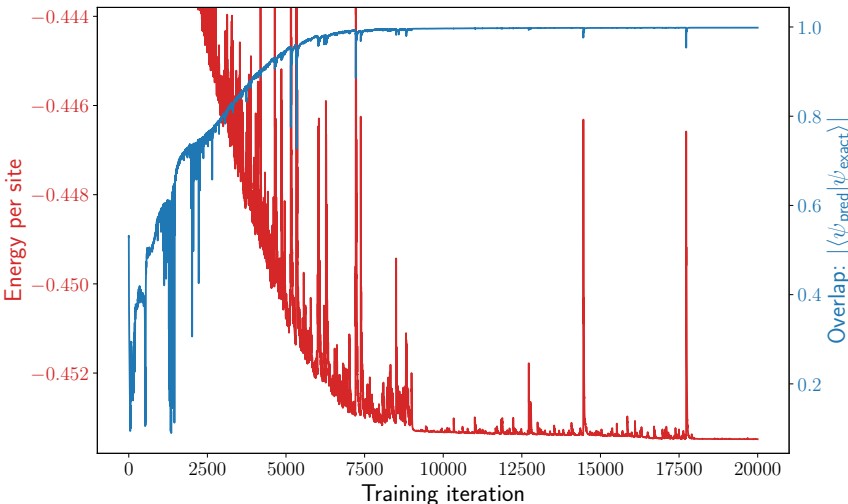

Figure 4: Training curve of energy and quantum state overlap for a kagome lattice with 12 nodes. This plot demonstrates the relationship between the improvement of energy and quantum state overlap, which validates that even a small improvement in energy can result in significant improvement in approaching the true ground state.

energies with GNN on the square lattice while falling behind on triangular and kagome lattices. For large systems at $J_2 \neq 0$, LCN outperforms GNN on all tested lattices, including square, honeycomb, and triangular. The good performance proves that the proposed LCN is able to accurately represent the ground states of quantum many-body systems without explicit structure encoding.

**Quantum State Overlap and Energy.** The quality of the variational result is measured by the ratio $(E - E_0)/\Delta$, where $\Delta$ is the gap of the Hamiltonian, i.e. the difference between the energies of the first excited state and the ground state. Usually, for the gapless model, the exact gap $\Delta$ is unknown. However, for the Heisenberg J1-J2 model, an approximate improvement on order $1/N^2$ on the energy per spin could be considered significant in approaching the true ground state. As listed in Table 1, our results can achieve this improvement in most cases. But even though energy value improvement is slightly less than order $1/N^2$, there still could be dramatic improvements in approaching the true ground state. We conduct an experiment to show this on a 12 nodes kagome lattice, where the true ground state can be obtained. As shown in Figure 4, we can observe that even though energy accuracy has 0.9% improvement (from -0.4492 to -0.4533), the ground state overlap (up to 1) can increase by 10% (from 0.9087 to 0.9971).

**Kernel Design Comparison.** Apart from applying regular convolution kernels on augmented lattices, we also design special convolution kernels for original honeycomb, triangular, and kagome lattices, which only capture the nearest neighbors of each center vertices. Details of these special kernel designs can be found at Appendix C.

We compare the performance of these two categories of kernel design on small systems over honeycomb, triangular, and kagome lattices, as shown in Table 2. Experimental results show that regular convolution kernels consistently outperform special kernels.

We argue the reasons are two folds. First, the regular kernel directly captures the interaction between a part of the second nearest neighbors, which is more helpful when $J_2$ is not zero. The second reason is that information reuse is hindered to some extent for special kernels. For example, when using special kernels for original kagome lattices, three distinctive kernel shapes need to be used for capturing different local structures on lattices. And the same local structure has little overlap with each other, resulting in less overlap between the same kernels. This impedes the information reuse among the same local structures that is crucial for capturing long-range spin correlations (Liang et al., 2018).

Table 2: Estimated ground state energy comparison between regular kernel and special kernel on honeycomb, triangular, and kagome lattices.

| Lattice | Size | $J_2$ | Regular Kernel | Special Kernel |
|---------|------|-------|----------------|----------------|
| Honeycomb | 32 | 0 | -0.5516(1) | -0.5512(2) |
|  |  | 0.2 | -0.4563(1) | -0.4547(3) |
| Triangular | 36 | 0 | -0.5601(4) | -0.5501(6) |
|  |  | 0.08 | -0.5273(4) | -0.5127(7) |
|  |  | 0.125 | -0.5126(6) | -0.4936(7) |
| Kagome | 36 | 0 | -0.4367(4) | -0.4207(6) |
|  |  | -0.02 | -0.4384(5) | -0.4253(6) |

So we conclude that lattice augmentation together with regular kernels is necessary for processing quantum lattice systems. On one hand, lattice augmentation is general for these four kinds of lattice and does not need any prior domain knowledge compared with GNN's sublattice encoding. On the other hand, with virtual vertices added, regular kernels can be applied, which has many advantages over special kernels such as boosting information exchange and reuse, as described in Section 3.2.

## 6 Conclusion

We propose lattice convolutions to process non-square lattices by converting them into grid-like augmented lattices through a set of operations. So regular convolution can be applied without using hand-crafted structure encoding, which is needed in the previous GNN method. And we design lattice convolutional networks that use self-gating and attention mechanisms to capture channel-wise interdependencies and spatial long-range spin correlations, which contribute to the high expressivity of variational wave functions. We experimentally demonstrate the effectiveness of lattice convolution network wave functions and achieve performance on par or better than existing methods.

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

## A    Summary of Test Lattices

Table 3: Test lattices settings with different lattice types, system sizes and $J_2$ values.

| Lattice | Number of Nodes | $J_2$ |
|---------|-----------------|-------|
| Square | 36 | 0, 0.5 |
|  | 100 | 0, 0.5 |
| Honeycomb | 32 | 0, 0.2 |
|  | 98 | 0, 0.2 |
| Triangular | 36 | 0, 0.08, 0.125 |
|  | 108 | 0, 0.125 |
| Kagome | 36 | 0, -0.02 |
|  | 108 | 0 |

## B    Implementation Detail

We use the same CNN architecture for all lattices. The convolution operation in each CNN layer is replaced with the proposed lattice convolution according to the lattice type. Pre-activation is not used for triangular lattices. For the kagome lattice of size 108, mask processing after convolution is not used.

We implement our models and VMC procedures in PyTorch (Paszke et al., 2019) and all models were trained on NVIDIA RTX A6000. For efficiency, we implement the VMC procedures with GPU parallelization. A batch of configurations are sampled in parallel at each step. Samples are kept at intervals to minimize the correlation between consecutive samples where the interval is equal to the size of the system. We further initialize each MCMC chain with a different random seed to maximize independence between chains. During training, we use the stochastic gradient estimated from each mini-batch of samples and optimize the models with the Adam optimizer (Kingma & Ba, 2015). The optimization generally converges within 30,000 steps. We save the model with the lowest stable training energy for testing, where the energies in neighboring steps have small variance. During testing, the energy is estimated from 200,000 configurations sampled from equilibrated Markov chains. Hyperparameters for optimization can be found in Appendix G. To estimate the stochastic error bars, we group Markov chains into 100 bins. The error bar is computed as the standard deviation of average energies from each bin.

## C    Special Kernel Design

We can design special convolution kernel structures based on different repetitive local patterns of different lattices, where each type of kernel only capture all the nearest neighbor nodes of the center vertices.

**Honeycomb.**    We consider the convolution kernel that covers the nearest neighbors. For every vertex in the honeycomb lattice, its nearest neighbors always form an equilateral triangle. However, for two adjacent vertices, the orientations of the triangles formed by their neighbors are different. Specifically, two neighbors can be related by a 180-degree rotation. As a result, as shown in Figure 5a, we rotate the convolution kernel by 180 degrees when going from one vertex to one of its neighbors.

**Triangular.**    For each center vertex, all of its nearest neighbors form a hexagon and this pattern is repeated across the whole lattice. So we can naturally design a hexagon shape convolution kernel to capture this pattern, as shown in Figure 5b. In practice, we implement this kernel by masking two positions of the square kernel at the diagonal corner.

**Kagome.**    We observe that the Kagome lattice has three different repetitive local patterns across the space, where each pattern can capture the nearest neighbors around the center vertices in the pattern. So we design

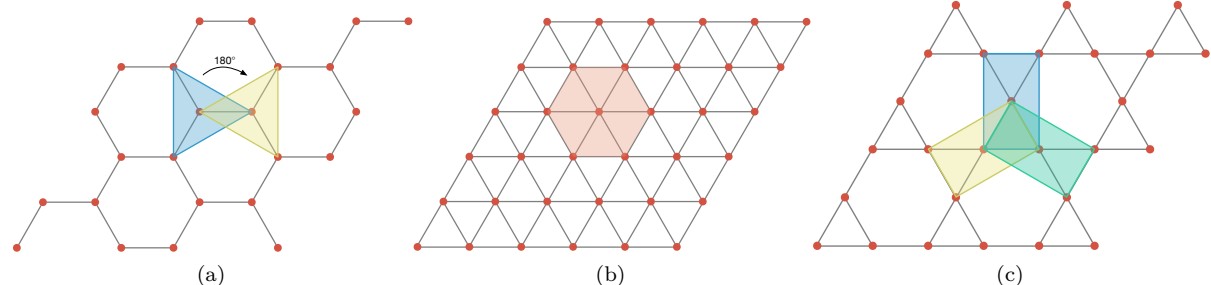

Figure 5: Special Kernel Design. (a) Honeycomb. (b) Triangular. (c) Kagome.

Table 4: Estimated ground state energy comparison between different kernel designs on the honeycomb lattice.

| Lattice | Size | $J_2$ | Regular Kernel | Special Kernel | 2-hop Special Kernel |
|---------|------|-------|----------------|----------------|----------------------|
| Honey-comb | 32 | 0 | -0.5516(1) | -0.5512(2) | -0.5511(2) |
|  |  | 0.2 | -0.4563(1) | -0.4547(3) | -0.4539(4) |

three different kernels corresponding to these three patterns, as shown in Figure 5c. And these three kernels do not share weights with each other.

**Different Kernels for Honeycomb.** For honeycomb lattices, the special kernel has half fewer parameters than the regular kernel. So for fairness comparison, we enlarge the special kernel to capture all next nearest neighbors and hence the number of parameters is similar to the regular kernel. As shown in Table 4, the performance of the 2-hop special kernel is still worse than the regular kernel with virtual vertices added. Another observation is that the 2-hop special kernel performs even slightly worse than the 1-hop special kernel, which implies that enlarging the receptive field of the special kernel cannot help capture further useful spin-spin correlation.

## D  Periodic Padding versus Zero Padding

As discussed in  Section 3.3, periodic padding is used before convolution in order to consider the periodic boundary condition for lattices. We compare the performance between periodic padding and zero padding, as shown in Table 5, which shows that periodic padding is essential for achieving better results.

Table 5: Estimated ground state energy comparison between periodic padding and zero padding.

| Lattice | Size | $J_2$ | Periodic padding | Zero padding |
|---------|------|-------|------------------|--------------|
| Honeycomb | 32 | 0 | -0.5516(1) | -0.5511(2) |
|  |  | 0.2 | -0.4563(1) | -0.4542(3) |
| Triangular | 36 | 0 | -0.5601(4) | -0.5109(6) |
|  |  | 0.08 | -0.5273(4) | -0.5095(5) |
|  |  | 0.125 | -0.5126(6) | -0.5091(5) |
| Kagome | 36 | 0 | -0.4367(4) | -0.4356(5) |
|  |  | -0.02 | -0.4384(5) | -0.4349(4) |

Table 6: Model parameters of Lattice Convolutional Networks and GNN. For the square lattice of size 100, the number for LCN is counted with 4 SE-Non-Local layers. We also use a 3-layer model which is slightly smaller and contains 1.0M parameters. See Appendix G for details.

| Lattice | Size | LCN (Ours) | GNN |
|---------|------|------------|-----|
| Square | 36 | **0.28M** | 0.86M |
| | 100 | 1.1M | 0.86M |
| Honeycomb | 32 | **0.36M** | 0.86M |
| | 98 | 1.9M | 0.86M |
| Triangular | 36 | **0.29M** | 0.86M |
| | 108 | 1.9M | 0.86M |
| Kagome | 36 | **0.49M** | 0.86M |
| | 108 | 4.6M | 0.86M |

## E  Model Capacity Comparison

In this section, we compare the number of model parameters between Lattice Convolutional Networks (LCN) and GNN (Kochkov et al., 2021) model over different lattices, as shown in Table 6. For triangular and kagome lattices with 36 nodes, LCN with only 0.29M and 0.49M trainable parameters is able to achieve better performance than the GNN model with 0.86M parameters. And on honeycomb and square lattices, LCN with 0.36M and 0.28M parameters yields the same results as the GNN model. So we argue that LCN is more expressive and parameter efficient than the GNN model on various small lattices.

As for the lattices with larger nodes, the number of parameters in LCN is increased, which is caused by the final MLP layer and boundary alignment operation on the augmented lattices. Since we flatten the final feature maps into a vector which is then processed by an MLP layer, the number of parameters will increase with the size of the network input. Also, we need to zero-pad the augmented lattices into parallelograms due to the irregular boundaries of the original lattice geometry structures, which then increases the size of the input feature map.

## F  Network Components

**Squeeze-and-Excitation Block.**  The process of SE-block can be represented as (Hu et al., 2018):

$$
\begin{aligned}
z_c = \mathbf{F}_{sq}\left(\mathbf{u}_c\right) &= \frac{1}{H \times W} \sum_{i=1}^{H} \sum_{j=1}^{W} u_c(i,j), \\
\mathbf{s} = \mathbf{F}_{ex}(\mathbf{z}, \mathbf{W}) &= \sigma(g(\mathbf{z}, \mathbf{W})) = \sigma\left(\mathbf{W}_2 \delta\left(\mathbf{W}_1 \mathbf{z}\right)\right), \\
\tilde{\mathbf{x}}_c = \mathbf{F}_{\text{scale}}\left(\mathbf{u}_c, s_c\right) &= s_c \mathbf{u}_c,
\end{aligned}
\tag{8}
$$

where $\mathbf{u}_c \in \mathbb{R}^{H \times W}$ denote the c-th output feature map of convolution operator. $z_c$ denote the c-th element of channel descriptor $\mathbf{z}$ squeezed from $\mathbf{u}_c$. $\delta$ refers to ReLU (Nair & Hinton, 2010) activation function and $\sigma$ refers to sigmoid function. $\mathbf{W}_1 \in \mathbb{R}^{\frac{C}{r} \times C}$ and $\mathbf{W}_2 \in \mathbb{R}^{C \times \frac{C}{r}}$. $r$ is the dimension reduction factor. The final output $\tilde{\mathbf{x}}_c$ is computed by channel-wise multiplication between $s_c$ and $\mathbf{u}_c$.

**Non-Local Block.**  Response at a position, denoted as $\mathbf{y}_i$, is computed as a weighted sum of features at all positions. The output of each position is formulated as (Wang et al., 2018):

$$
\begin{aligned}
f\left(\mathbf{x}_i, \mathbf{x}_j\right) &= e^{\theta(\mathbf{x}_i)^T \phi(\mathbf{x}_j)}, \\
\mathbf{y}_i &= \frac{1}{\mathcal{C}(\mathbf{x})} \sum_{\forall j} f\left(\mathbf{x}_i, \mathbf{x}_j\right) g\left(\mathbf{x}_j\right), \\
\mathbf{z}_i &= W_z \mathbf{y}_i + \mathbf{x}_i,
\end{aligned}
\tag{9}
$$

where $g$ is a linear embedding that transform the input feature map $\mathbf{x}$: $g(\mathbf{x}_i) = W_g \mathbf{x}_i$. And $f(\mathbf{x}_i, \mathbf{x}_j)$ is embedded gaussian. $\theta(\mathbf{x}_i) = W_\theta \mathbf{x}_i$ and $\phi(\mathbf{x}_j) = W_\phi \mathbf{x}_j$ are two embeddings to compute dot product similarity. $\mathcal{C}(\mathbf{x}) = \sum_{\forall j} f(\mathbf{x}_i, \mathbf{x}_j)$ is normalization factor. $\mathbf{z}_i$ is the final output of this block at position $i$ by adding residual connection.

## G Hyperparameters

Table 7: Hyperparameters for optimization. The learning rate is multiplied by 0.1 at every decay step. During training, the gradient norm is clipped from the beginning for square and honeycomb lattices. For triangular and kagome lattices, the gradient norm is clipped after the first learning rate decay. For all experiments, we use a 3x3 convolution kernel and 64 output channels for each layer.

| Lattice | Size | $J_2$ | Number of SE-Non-Local layers | Batch size | Learning rate | Learning rate decay steps | Gradient norm clip |
|---|---|---|---|---|---|---|---|
| Square | 36 | 0 | 2 | 500 | 1e-3 | 20000, 40000, 60000 | 1 |
| | | 0.5 | 2 | 500 | 1e-3 | 20000, 40000, 60000 | 1 |
| | 100 | 0 | 3 | 200 | 5e-4 | 8000, 12000, 16000 | 1 |
| | | 0.5 | 4 | 200 | 5e-4 | 8000, 12000, 16000 | 1 |
| Honeycomb | 32 | 0 | 2 | 500 | 1e-3 | 20000, 40000, 60000 | 1 |
| | | 0.2 | 2 | 500 | 1e-3 | 20000, 40000, 60000 | 1 |
| | 98 | 0 | 4 | 100 | 7e-4 | 8000, 12000, 16000 | 1 |
| | | 0.2 | 4 | 100 | 7e-4 | 10000, 16000, 22000 | 1 |
| Triangular | 36 | 0 | 2 | 1000 | 1e-3 | every 4000 | 2 |
| | | 0.08 | 2 | 1000 | 1e-3 | every 4000 | 2 |
| | | 0.125 | 2 | 1000 | 1e-3 | every 4000 | 2 |
| | 108 | 0 | 2 | 200 | 1e-3 | every 4000 | None |
| | | 0.125 | 2 | 200 | 1e-3 | every 4000 | None |
| Kagome | 36 | 0 | 2 | 1000 | 1e-3 | every 4000 | 1 |
| | | -0.02 | 2 | 1000 | 1e-3 | every 4000 | 1 |
| | 108 | 0 | 2 | 200 | 1e-3 | every 4000 | 1 |

## H Ablation study

### H.1 Mask Operation

As described in Section 3.3, we reset all vertices used for the boundary alignment and the periodic padding to zero after each convolution. The vertices used for boundary alignment do not belong to the original lattice area and periodic padding, they are only used for transforming the original lattice area to regular shape where regular square kernel can be applied to the boundary. Also, the vertices used for periodic padding do not belong to the original lattice area. So this motivates us to use mask operation. But in practice we found mask operation could be an optional choice but it indeed has benefits in certain cases. For example, mask operation could improve performance on a kagome lattice of size 36, as shown in Table 8.

Table 8: Estimated ground state energy comparison between the mask and non-mask operation on the Kagome lattice.

| Lattice | Size | $J_2$ | Mask | No Mask |
|---|---|---|---|---|
| Kagome | 36 | 0 | -0.4367(4) | -0.4364(6) |
| | | -0.02 | -0.4384(5) | -0.4336(4) |

### H.2 Virtual Vertices in Feature Passing

As described in Section 3.2, during convolution, the virtual vertices participate in the convolution in the same way as the original vertices. i.e., values of virtual vertices are also updated. Virtual vertices can help

boost information exchange by participating in feature passing. To demonstrate the effectiveness of including virtual vertices in convolution, we perform an ablation study on whether to use virtual vertices in feature passing. By resetting the virtual vertices value to zero after each convolution, we exclude the virtual vertices from feature passing. The result is shown in Table 9, and we can see that including virtual vertices in the convolution can improve the performance.

Table 9: Estimated ground state energy comparison between whether including virtual vertices (VV) in convolution or not on the Kagome lattice, *i.e.*, whether reset virtual vertices to zero after each convolution.

| Lattice | Size | $J_2$ | No VV reset | VV reset |
|---------|------|-------|-------------|----------|
| Kagome  | 36   | 0     | -0.4367(4)  | -0.4321(5) |
|         |      | -0.02 | -0.4384(5)  | -0.4354(5) |

### H.3 Network Architecture Comparison

We conduct experiments on the kagome lattice to show the effect of the Squeeze-and-Excitation (SE) block and Nonlocal block, as shown in Table 10. Adding an SE block will improve performance upon using residual connection only. We hypothesize that SE blocks use squeeze operation to aggregate global spatial information and rescaling to recalibrate the importance of channels to capture channel-wise dependencies, which can implicitly capture long-range spin correlation. And non-local block makes spin at one position attend to all other spins, which explicitly capture spin-spin global interaction, so using the non-local block can further improve performance.

## I Full Range J2 on Square Lattice

For completeness, we compare with other physics-incorporated methods targeting square lattice. We conduct experiments for full range J2 on a square lattice of size 36, as shown in Table 11. Compared with CNN (Choo et al., 2019), our method consistently performs well on the full range of J2, especially for the frustrated regime ($J2 \approx 0.5$) and small J2 while without using any prior physical knowledge. For large J2, CNN is better than ours, it might be because they enforce $C_4$ symmetry which is important in the striped order phase at large J2 (Choo et al., 2019). Besides, CNN incorporates the Marshall sign rule which defines the sign structure of the ground state wavefunction for square lattice at J1=0 or J2=0. Marshal sign rule only works for bipartite graphs (such as square lattice) and non-frustrated regimes. If the prior sign rule is violated (J2=0.5), the CNN result is worse than ours, as expected.

The latest SOTA methods RBM+PP (Nomura & Imada, 2021) and RBM+Lanczos (Chen et al., 2022) have very good results on the square lattice. RBM+PP is a hybrid method that heavily relies on intense physical knowledge (GNN (Kochkov et al., 2021) uses some results from RBM+PP as reference energy). RBM+PP combines the restricted Boltzmann machine and the pair-product wavefunction. The pair-product wavefunction has been used in physics for a long time and it is known that the pair-product wavefunction excellently captures the ground state of the Heisenberg model on a square lattice. Therefore the combination RMB+PP is very effective on square lattices. However, it is unclear how it performs on other lattices. RBM+Lanczos incorporates spin flip, translation, and lattice point group symmetries for square lattice in the network. And it uses Lanczos iterations to further improve results. But the computation cost of the Lanczos method increases dramatically with Lanczos steps, and Lanczos correction will become smaller for larger systems.

## J GNN with and without Sublattice Encoding.

To illustrate the necessity of using hand-crafted sublattice encoding in the GNN model, we try to implement the GNN model based on details provided in Kochkov et al. (2021) and reproduce the results. We conduct an ablation study on sublattice encoding and summarize the results in Table 12. For triangular and kagome lattices, we cannot get valid results without using sublattice encoding (either too large or too small). One

Table 10: Estimated ground state energy comparison between different network architecture choices. We use the Kagome lattice as an example to show the effectiveness of adding the Squeeze-and-Excitation (SE) block and the Nonlocal block to the network.

| Lattice | Size | $J_2$ | Residual only | Residual+SE | Residual+Nonlocal | Residual+SE+Nonlocal |
|---------|------|-------|---------------|-------------|-------------------|----------------------|
| Kagome | 36 | 0 | -0.4310(5) | -0.4323(5) | -0.4365(5) | -0.4367(4) |
|        | 108 | 0 | -0.4090(5) | -0.4153(6) | -0.4161(5) | -0.4276(3) |

Table 11: Estimated ground state energy for full range J2 on 36 sites square lattice. Results for exact energies and CNN are taken from Choo et al. (2019). Results for RBM+PP are taken from Nomura & Imada (2021). Results for RBM+Lanczos are taken from (Chen et al., 2022) (only $J_2 = 0.5, 0.55, 0.6$ use one Lanczos step). Our method is the only one without any prior physical knowledge incorporated in the model.

| Square 6x6 | $J_2 = 0$ | $J_2 = 0.2$ | $J_2 = 0.4$ | $J_2 = 0.45$ | $J_2 = 0.5$ | $J_2 = 0.55$ | $J_2 = 0.6$ | $J_2 = 0.8$ | $J_2 = 1$ |
|------------|-----------|-------------|-------------|--------------|-------------|--------------|-------------|-------------|-----------|
| Exact (Schulz et al., 1996) | -0.678872 | -0.599046 | -0.529745 | - | -0.503810 | -0.495178 | -0.493239 | -0.586487 | -0.714360 |
| CNN (Choo et al., 2019) | -0.67882(1) | -0.59895(1) | -0.52936(1) | -0.51452(1) | -0.50185(1) | -0.49067(2) | -0.49023(1) | -0.58590(1) | -0.71351(1) |
| RBM+Lanczos (Chen et al., 2022) | -0.678868(2) | -0. 599044(3) | -0.529687(7) | -0.51552(1) | -0.50376(3) | -0.49512(4) | -0.49313(5) | -0.586411(9) | -0.71429(1) |
| RBM+PP (Nomura & Imada, 2021) | - | - | -0.529726(1) | -0.515633(1) | -0.503765(1) | -0.495075(1) | - | - | - |
| LCN (ours) | -0.6788(1) | -0.5990(1) | -0.5295(1) | -0.5151(1) | -0.5022(2) | -0.4910(2) | -0.4840(2) | -0.5855(2) | -0.7119(3) |

possible reason is that the distribution of learned wave function is very hard for accurate sampling. On square and honeycomb lattices, sublattice encoding is essential for achieving better results. So we can draw the conclusion that the hand-crafted sublattice encoding is important for the GNN model.

## K    Results Comparison with GNN-2.

In Kochkov et al. (2021), they also provide another variant of GNN ansatz called GNN-2 that adopts two identical GNN models to predict the amplitude and argument of wave functions separately. And these two GNN models do not share weights. Since GNN-2 doubles the parameter of GNN ansatz and predicts the amplitude and argument each with a GNN model, so GNN-2 is more expressive and performs better than GNN. The comparison between LCN and GNN-2 is presented in Table 13. Note that although LCN uses a single branch network to simultaneously predict the amplitude and argument of wave functions, similar to GNN, it still shows competitive performance with the GNN-2 model.

Table 12: Reproduced GNN performance with and without sublattice encoding.

| Lattice | Size | $J_2$ | GNN with sublattice | GNN without sublattice |
|---------|------|-------|---------------------|------------------------|
| Square | 36 | 0.0 | -0.6776(6) | -0.6536(11) |
| Honeycomb | 32 | 0.2 | -0.4551(3) | -0.4498(48) |
| Triangular | 36 | 0.0 | -0.5571(5) | N/A |
| | | 0.08 | -0.5253(5) | N/A |
| | | 0.125 | -0.5119(6) | N/A |
| Kagome | 36 | 0.0 | -0.4345(5) | N/A |

Table 13: Estimated energy per site of the learned wave function (with error bars in parenthesis). Lower is better. Four kinds of lattices with various sizes are used for comparison. The $J_2$ value controls the next nearest neighboring interaction on the lattice, resulting in differences in the ground states. The best result in each row is denoted in bold. We use $*$ to denote the results measured from plots in their reference papers. For the reference ground state energies, we also annotate the employed methods: † for exact diagonalization, $\infty$ for infinite-size estimates, ‡ for RMB+PP, § for QMC methods, and ¶ for DMRG methods.

| Lattice | Size | $J_2$ | GNN-2 | LCN (ours) | Reference Energy |
|---------|------|-------|-------|-----------|------------------|
| Square | 36 | 0 | - | **-0.6788(1)** | -0.678872† |
| | | 0.5 | **-0.5023(5)** | -0.5022(2) | -0.503810† |
| | 100 | 0 | - | **-0.6708(1)** | 0.671549(4)§ |
| | | 0.5 | **-0.4960(5)** | -0.4957(4) | -0.497629(1)‡ |
| Honeycomb | 32 | 0 | - | **-0.5516(1)** | -0.5517†* |
| | | 0.2 | **-0.4564(7)** | -0.4563(1) | -0.4567†* |
| | 98 | 0 | - | **-0.5421(4)** | -0.5448∞* |
| | | 0.2 | -0.4536(5) | **-0.4538(4)** | -0.4527∞* |
| Triangular | 36 | 0 | -0.55889 | **-0.5601(4)** | -0.5603734† |
| | | 0.08 | - | **-0.5273(4)** | -0.5286†* |
| | | 0.125 | **-0.5131(8)** | -0.5126(6) | -0.515564† |
| | 108 | 0 | **-0.5519(4)** | -0.5475(4) | -0.551∞ |
| | | 0.125 | -0.5069(8) | **-0.5110(4)** | -0.5126¶ |
| Kagome | 36 | 0 | -0.4338(6) | **-0.4367(4)** | -0.43837653† |
| | | -0.02 | - | **-0.4384(5)** | -0.4399†* |
| | 108 | 0 | **-0.4314(7)** | -0.4276(3) | -0.4386¶ |

