# OpenReview forum: "Lattice Convolutional Networks for Learning Ground States of Quantum Many-Body Systems"
_TMLR — Rejected by TMLR_

### Review · Reviewer_x5EQ · 2023-06-17

**Summary Of Contributions:**

The authors present a novel approach for using deep neural networks to learn the ground energy of spin systems. Compared with prior deep learning approaches, the proposed method can be used to learn more spin systems with lattice structures. The key idea is the augmentation method to transform honeycomb, kagome lattices, and triangular lattices can be transformed to a squared structure, which then can be manipulated by convolutional neural networks (CNNs). Extensive experimental results demonstrate the effectiveness of the proposed method.  Overall, this work contributes to the advancement of deep learning techniques in the realm of quantum systems.

**Audience:**

Yes

**Broader Impact Concerns:**

NA.

**Claims And Evidence:**

No

**Requested Changes:**

The issues mentioned in 'Strengths And Weaknesses' should be addressed.

In addition, Eq. equation 1 on Page 3 should be corrected.

In Equation (1), \psi(c_i) should be the probability amplitude of the configuration c_i.

**Strengths And Weaknesses:**

Exploring the potential of deep learning in advancing the study of quantum systems is a key topic at the intersection of quantum physics and computer science. The manuscript presents intriguing findings in this area; however, several aspects require further clarification.

The proposed augmentation method, although novel in the context of non-square spin systems, has been extensively discussed in the deep learning literature. For instance, the work by Hu et al. [Hu, Shi-Min, et al. "Subdivision-based mesh convolution networks." ACM Transactions on Graphics (TOG) 41.3 (2022): 1-16.] introduces a similar CNN framework capable of learning 3D triangle meshes. Therefore, it is important to highlight the specific contributions and differentiating factors of the proposed method.

Apart from the augmentation method, it is challenging to identify significant contributions in the submission. While the authors demonstrate a marginal improvement of L-CNN over GNN in learning the Heisenberg model, it is worth noting that neither of these approaches represents the mainstream methods for investigating quantum systems. Conventional techniques like Matrix Product State (MPS), Density Matrix Renormalization Group (DMRG), and deep neural ansatz face difficulties due to the exponentially large quantum state space. Alternatively, when applying these methods to learn quantum systems, a crucial step involves leveraging the Hamiltonian's symmetry to design an ansatz capable of accurately expressing the desired states using a polynomial or even a small number of parameters. Notable work in this regard is presented by Favoni et al. [Favoni, Matteo, et al. "Lattice gauge equivariant convolutional neural networks." Physical Review Letters 128.3 (2022): 032003.].

In comparison with DMRG or MPS, it is essential to explore the tradeoff between precision and runtime. Although the authors demonstrate that L-CNN exhibits a slight advantage over DMRG, it is important to conduct a more comprehensive comparison that considers both the precision of results and the computational time required. Such an analysis would provide a clearer understanding of the advantages offered by the proposed method.

---

> ### Author Response · Authors · 2023-07-01
> **Response to Reviewer x5EQ -- Part 1**
>
> Thank you for your time and valuable comments! We hope your concerns and questions can be addressed by our responses below.
>
> >**Q1: The proposed augmentation method, although novel in the context of non-square spin systems, has been extensively discussed in the deep learning literature. For instance, the work by Hu et al. [Hu, Shi-Min, et al. "Subdivision-based mesh convolution networks." ACM Transactions on Graphics (TOG) 41.3 (2022): 1-16.] introduces a similar CNN framework capable of learning 3D triangle meshes. Therefore, it is important to highlight the specific contributions and differentiating factors of the proposed method.**
>
> - Thank you for pointing out that paper. Apart from SubdivNet [1], we also noticed another representative paper on learning triangle meshes, known as MeshCNN [2]. In the revision, we cited these papers in the "Relationship with Prior Work" part and highlight the difference with our work. These work focus on how to harness the potential of CNN on irregular structures, particularly 3D triangle meshes. These work utilize the unique properties of triangular meshes and design specialized convolutional neural networks that operate on the meshes by leveraging their intrinsic connections. Specifically, SubdivNet [1] perform CNN operations on 3D triangle meshes using Loop subdivision sequence connectivity and make an analogy between mesh faces and pixels in a 2D image, which leads to a new mesh convolution operator that can aggregate local features from nearby faces. MeshCNN [2] observed that the edges of a mesh are analogous to pixels in an image, as each edge has exactly four near edges in the one-ring neighborhood so CNN can be applied in this sense.
>
> - Our method distinguishes with these works in several ways: (1) We aim to solve ground state of spin systems instead of 3D shape analysis, and we want to apply CNN on different kinds of lattices instead of triangle meshes. Different lattices have different local topologies and sometimes there are multiple local patterns even for the same lattice, such as Kagome lattice. So we need to have a principled way to process these lattices. Thus, SubdivNet or MeshCNN method aren't directly applicable to quantum many-body problems on lattice systems. (2) From the image analogy perspective, SubdivNet treats mesh faces as pixels, MeshCNN treats mesh edges as pixels, whereas our method treats lattice points and virtual vertices as pixels. (3) Using virtual vertices to augment original lattice is not just based on geometry observation but features physical motivation for solving quantum many-body problem, which will be elaborated in the following.
>
> - About the third point mentioned above, apart from enabling traditional CNN to be used, introducing virtual vertices provides other advantages specific to solving the quantum many-body problem. The benefit of adding virtual vertices are two folds as mentioned in the last paragraph of Section 3.2. First, it can gather and distribute the information from the original vertices, which can help increase receptive fields and boost information exchange. Second, we can overlap the same convolution kernel in order to enable information reuse, which is crucial to capture long-range spin correlations [3][4]. In [3], it bridges the convolutional neural network to the tensor network states and proposes that information reuse introduced by overlapping convolution filters can enhance the state representation ability. Our experiment on comparing regular kernel and special kernel in Table 2 can also somewhat validate this. Specifically, if we don't use virtual vertices to augment the original lattice, we need to design special kernels tailored to different local structures of different kinds of lattices, which performs worse than regular kernel (virtual vertices are used) and part of reason is less overlap among same kernels. For example, when using special kernels for original kagome lattices, three distinctive kernel shapes need to be used for capturing different local structures on lattices. And the same local structure has little overlap with each other, which results in less overlap within the shared kernels. To sum, adding virtual vertices to augment original lattices has physical motivation not just only for applying CNN to irregular structures.
>
>
> [1] Hu, Shi-Min, et al. "Subdivision-based mesh convolution networks." ACM Transactions on Graphics (TOG) 41.3 (2022): 1-16.\
> [2] Hanocka, Rana, et al. "Meshcnn: a network with an edge." ACM Transactions on Graphics (TOG) 38.4 (2019): 1-12.\
> [3] Y. Levine, O. Sharir, N. Cohen and A. Shashua, Bridging Many-Body Quantum Physics and Deep Learning via Tensor Networks, arXiv: 1803.09780v1.\
> [4] Xiao Liang, Wen-Yuan Liu, Pei-Ze Lin, Guang-Can Guo, Yong-Sheng Zhang, and Lixin He. Solving frustrated quantum many-particle models with convolutional neural networks. Physical Review B, 98(10):104426, 2018.\

---

> ### Author Response · Authors · 2023-07-01
> **Response to Reviewer x5EQ -- Part 2**
>
> >**Q2: Apart from the augmentation method, it is challenging to identify significant contributions in the submission. While the authors demonstrate a marginal improvement of L-CNN over GNN in learning the Heisenberg model, it is worth noting that neither of these approaches represents the mainstream methods for investigating quantum systems. Conventional techniques like Matrix Product State (MPS), Density Matrix Renormalization Group (DMRG), and deep neural ansatz face difficulties due to the exponentially large quantum state space. Alternatively, when applying these methods to learn quantum systems, a crucial step involves leveraging the Hamiltonian's symmetry to design an ansatz capable of accurately expressing the desired states using a polynomial or even a small number of parameters. Notable work in this regard is presented by Favoni et al. [Favoni, Matteo, et al. "Lattice gauge equivariant convolutional neural networks." Physical Review Letters 128.3 (2022): 032003.].**
>
> - Thanks for the question. DMRG and MPS are very powerful for 1D and quasi-1D quantum systems because of the low entanglement of the ground state, but not for 2D systems. This limitation motivates using deep neural anstaz for 2D systems. Earlier study used restricted Boltzmann machines (RBMs), but RBM suffers from highly entangled quantum systems. Then other deep neural anstaz, such as CNN, are used to address this problem. But CNN does not directly generalize to non-square and more complicated lattices. Therefore, we introduce LCN for this problem. It is true that LCN and GNN are not the conventional methods, but they exemplify the innovative use of deep neural networks as an ansatz to obtain the ground state of more complex 2D quantum spin systems.
>
> - GNN [1] holds the distinction of being the pioneering deep neural network ansatz adaptable to a range of prevalent lattices, including square, triangular, honeycomb, and kagome. Similarly, our work is geared towards solving the ground state of quantum spin systems across various lattice types. Consequently, drawing a comparison with the GNN model is logical.  (Continue on Part 3)

---

> ### Author Response · Authors · 2023-07-01
> **Response to Reviewer x5EQ -- Part 3**
>
> - In addition, we respectfully disagree with the notion that the simplicity of our method negates its contribution. In fact, through incisive observations of various lattices, we have devised a simple, elegant, and effective solution for addressing the quantum many-body problem across a spectrum of lattices. Our key contributions and their ramifications are outlined as follows:
>   - Our method can effectively learn ground state of quantum systems on multiple kinds of lattices. This stands in contrast to the majority of neural quantum state studies, which are predominantly confined to square lattices.
>   - We have introduced a simple yet principled approach for handling non-square lattices. Our research elucidates the capability of convolutional neural networks to adeptly and accurately learn the ground state within these non-square lattices.
>   - A notable aspect of our approach is its independence from any prior physical knowledge, and yet it delivers performance that is either on par or surpasses that of the existing solution [1] that incorporating physical knowledge. This is particularly invaluable in situations where there is a lack of prior knowledge.
>   - Our work has achieved good results, particularly on the kagome lattice, where the relative error is at most four times smaller compared to GNN [1]. This is especially noteworthy given that the neural quantum state field has seldom considered the kagome lattice. Understanding the ground state's property of the Heisenberg model on Kagome lattice is one of the main focuses of condensed matter research. However, the ground state of the Kagome Heisenberg model is notoriously difficult to study.
>
> - Regarding symmetries in Hamiltonian, our LCN does not incorporate these symmetries directly within the network. This is an intentional design choice as our objective is to construct a method that is not only simplistic and potent but also universal, capable of being applied to an assortment of lattices without reliance on any prior knowledge. This attribute is particularly advantageous in scenarios where such knowledge is absent. Consequently, our method boasts a broader scope of applicability.
>
> - But we agree that symmetry is important in physics, and we have acknowledged as much by citing the relevant paper [2] in the revision. However, it's pertinent to note that one of the objectives in our current work is to develop a neural network ansatz that operates independently of physical knowledge.  With this in mind, the exploration of symmetries will be left for future research endeavors.
>
>
> [1] Dmitrii Kochkov, Tobias Pfaff, Alvaro Sanchez-Gonzalez, Peter Battaglia, and Bryan K Clark. Learning ground states of quantum hamiltonians with graph networks. arXiv preprint arXiv:2110.06390, 2021.\
> [2] Favoni, Matteo, et al. "Lattice gauge equivariant convolutional neural networks." Physical Review Letters 128.3 (2022): 032003.
>
> >**Q3: In comparison with DMRG or MPS, it is essential to explore the tradeoff between precision and runtime. Although the authors demonstrate that L-CNN exhibits a slight advantage over DMRG, it is important to conduct a more comprehensive comparison that considers both the precision of results and the computational time required. Such an analysis would provide a clearer understanding of the advantages offered by the proposed method.**
>
> - Similar to answer in Q2, DMRG or MPS have been well-established and have proven extremely successful for certain classes of problems, especially one-dimensional quantum systems with limited entanglement. For those problem that DMRG or MPS can perfectly solve, they are indeed much faster than deep neural ansatz as they don't need computationally costly MCMC sampling. However, DMRG and MPS have limitations in capturing high entanglement in 2D systems. Deep neural networks, with their inherent flexibility and capacity to represent intricate functions, are adept at learning correlations between variables. Consequently, they possess the capabilities to more naturally represent the complex correlations and entanglements found in many-body quantum states, in contrast to MPS. Therefore, deep neural ansatz emerges as a promising alternative to MPS and DMRG for 2D systems.
>
> >**Q4: In addition, Eq. equation 1 on Page 3 should be corrected. In Equation (1), \psi(c_i) should be the probability amplitude of the configuration c_i.**
>
> - Thanks for pointing out this. $\psi(c_i)$ is indeed the probability amplitude of the configuration $c_i$, and also the wavefunction value of $c_i$. We modified this sentence in the revision.

---

### Review · Reviewer_f671 · 2023-06-18

**Summary Of Contributions:**

The paper proposes a method to apply CNNs to estimate ground states of quantum systems, but on non-rectangular grids. The lattices they do consider are structured but are not standard grid-like. The authors show that such an approach is better than using GNNs to handle the non-square grids. The authors compare their results to other methods compatible with non-square grids such as GNNs and show that their method is competitive.

**Audience:**

Yes

**Broader Impact Concerns:**

N/A.

**Claims And Evidence:**

No

**Requested Changes:**

Show the benefits of the approach compared to existing methods. I get and believe that these kinds of CNNs are better than GNNs for the proposed lattices. But - this also should be demonstrated time-wise, as the accuracy difference is quite marginal.


I would like to see a time comparison with standard methods like Lanczos (+ preconditioning, if possible), implemented on the same GPU (or, let the CNN model run on CPU for a fair comparison).


**Strengths And Weaknesses:**

Strengths:

The examples that the authors consider are non-square but are structured. Given this setup, it is a good idea to adapt a CNN to such cases rather than use a GNN, both in terms of capacity and in terms of performance.

Weaknesses:

1) I am not familiar with this field, so maybe the confusion comes from there, but I do not understand several aspects of the experimental setup. For example: do the authors always consider linear problems that can be solved using an eigenvalue decomposition? If that’s the case, what is the advantage of using NNs (CNNs or GNNs) to solve such problems? Is it simply faster?


2) Continuing on the previous question. What is the benefit of using the proposed method? The energies look quite comparable between the columns of Table 1, so I guess that the issue here is time, as one can always solve an eigenproblem. But, the authors do not show this at all in the experiments.


3) Figure 3: it is not clear what is the input and what is the output of the network. Is x just a point in space? Is it the lattice itself? And the output size?


4) As far as I understand, there is no generalization in this work. So, if the problem is solved on one grid-type, the same network’s weights are not usable for another grid type, and the whole network needs to be retrained. However, training is costly. Furthermore, the training includes multiple matrix-vector products with $H$, so the training is more or less the same as a Lanczos process. What is the benefit then? Aren’t there classical preconditioning techniques to accelerate the Lanczos algorithm? How does the method compare to them?


5) GNNs can work with arbitrary domains, while here the domains are non-square but are highly structured. So – there is an advantage of GNNs. It is not clear how common are these problems in nature. Is there a lot of interest in such cases? How about unstructured meshes (where, say, the density of the triangles changes across the domain like Finite Element examples)?


6) It is not clear what is “size” in Table 1. Is it N? So x is in 2^N?


7) Based on the citations, it seems that this paper should go to a Physics venue, as the method is quite straightforward in terms of the ML aspects. The method converts a GNN to a CNN-type computation on a non-square but structured grid, and I am sure this was done before in a simpler vision task. Therefore, the methodology is quite straightforward aside from this application, which is relevant mostly to the Physics audience as the citations hint. This is not a major issue, though.

---

> ### Author Response · Authors · 2023-07-02
> **Response to Reviewer f671 -- Part 1**
>
> Thank you for your time and valuable comments! We hope your concerns and questions can be addressed by our responses below.
>
> >**Q1: I am not familiar with this field, so maybe the confusion comes from there, but I do not understand several aspects of the experimental setup. For example: do the authors always consider linear problems that can be solved using an eigenvalue decomposition? If that’s the case, what is the advantage of using NNs (CNNs or GNNs) to solve such problems? Is it simply faster?**
>
> - Sorry for causing the confusion. We need to clarify some misunderstanding about the quantum many-body problem first:
>   - There's no simplification in our problem setting. Finding the ground state of a quantum system is equivalent to solving a particular type of eigenvalue problem. More specifically, it involves solving the Schrödinger equation, which is an eigenvalue equation.
>
>   - In quantum mechanics, the state of a system is described by a wavefunction, which is a solution to the Schrödinger equation. This equation includes an operator called the Hamiltonian, which represents the total energy of the system. The eigenvalues of this equation represent the possible energies the system can have, and the corresponding eigenfunctions (the wavefunctions) describe the states of the system associated with these energies.
>
>   - The ground state of a quantum system is the state with the lowest possible energy. This means that when you solve the Schrödinger equation for the system, the ground state corresponds to the lowest eigenvalue and the corresponding eigenfunction.
>
> - About why this problem cannot always be solved directly using eigenvalue decomposition:
>   - While finding the ground state of a quantum system does indeed involve solving an eigenvalue problem, using straightforward eigenvalue decomposition is not always feasible. For many-body quantum systems, the size of the Hamiltonian matrix grows exponentially with the number of particles in the system. This "curse of dimensionality" makes it computationally intractable to directly compute the eigenvalues and eigenvectors of the Hamiltonian matrix for anything but the smallest systems. For example, the size of the Hamiltonian matrix for a 10x10 square lattice is $2^{100} \times 2^{100}$, which is impossible to solve directly or even stored in any current or foreseeable computer.
>   - Neural networks can parameterize the wavefunction itself instead of explicitly obtaining the whole state vector such as in exact diagonalization, which is memory efficient. Moreover, neural networks are universal function approximators, which means they have the theoretical ability to represent any function to arbitrary accuracy. This is a desirable feature for representing high-dimensional wavefunctions, and partially mitigating the "curse of dimensionality". In addition, neural networks are highly flexible and can approximate a wide variety of functional forms, which also makes them potentially well-suited to represent the ground state wavefunction of complex quantum systems.

---

> ### Author Response · Authors · 2023-07-02
> **Response to Reviewer f671 -- Part 2**
>
> >**Q2: Continuing on the previous question. What is the benefit of using the proposed method? The energies look quite comparable between the columns of Table 1, so I guess that the issue here is time, as one can always solve an eigenproblem. But, the authors do not show this at all in the experiments.**
>
> - First of all, we would like to clarify that the accuracy difference actually is not marginal. The objective of solving quantum many-body problem is to obtain the ground state of the system, and we can only use energy to evaluate how accurate the solved ground state is for large systems. As described in Section 5 "Quantum State Overlap and Energy" part, for the J1-J2 Heisenberg model, approximately improvement on order $1/N^2$ on the energy per spin could be considered significant in approaching the true ground state, where $N$ is the size of the system. And our results can achieve this improvement in most cases. In addition, we present a case study involving a 12 nodes kagome lattice, for which the true ground state can be obtained by exact diagonalization, to illustrate that there could be significant improvements in approaching the true ground state even though the corresponding energy value improvement appears to be marginal. For example, as shown in Figure 4, we observe that even though energy accuracy has a 0.9% improvement (from -0.4492 to -0.4533), which is less than $1/N^2$, the ground state overlap (up to 1) can increase by 10% (from 0.9087 to 0.9971), which is a significant improvement.
>
> - Second, similar to answer in Q1, while finding the ground state of a quantum system does involve solving an eigenvalue problem, directly using eigenvalue decomposition is not always feasible. For many-body quantum systems, the size of the Hamiltonian matrix grows exponentially with the number of particles in the system. This "curse of dimensionality" makes it computationally intractable to directly compute the eigenvalues and eigenvectors of the Hamiltonian matrix for anything but the smallest systems.
>
> - For the time comparison with GNN [1], since GNN doesn't release code, we tried our best to implement GNN to reproduce their results. Computational complexity of variational Monte Carlo is largely determined by the efficiency of evaluating the wave-function amplitudes during MCMC sampling. We test the efficiency on a 36 nodes triangular lattice with a single NVIDIA 2080Ti GPU, and our observation is that for each MCMC step, GNN is ~10 $\times$ slower (GNN is ~4s, ours is ~0.4s) than our method.
>
> [1] Dmitrii Kochkov, Tobias Pfaff, Alvaro Sanchez-Gonzalez, Peter Battaglia, and Bryan K Clark. Learning ground states of quantum hamiltonians with graph networks. arXiv preprint arXiv:2110.06390, 2021.
>
> >**Q3: Figure 3: it is not clear what is the input and what is the output of the network. Is x just a point in space? Is it the lattice itself? And the output size?**
>
> - For a wavefunction, input is a specific spin configuration $c$, and output is the wavefunction value $\psi(c)$ (a complex number). The network can be seen as the wavefunction itself. So the network input is also a specific spin configuration on the lattice, and the output is the amplitude and phase of the wavefunction value (since we use real-valued network) associated with that spin configuration. We revised the caption of Fig 3 to make it more clear.

---

> ### Author Response · Authors · 2023-07-02
> **Response to Reviewer f671 -- Part 3**
>
> >**Q4: As far as I understand, there is no generalization in this work. So, if the problem is solved on one grid-type, the same network’s weights are not usable for another grid type, and the whole network needs to be retrained. However, training is costly. Furthermore, the training includes multiple matrix-vector products with, so the training is more or less the same as a Lanczos process. What is the benefit then? Aren’t there classical preconditioning techniques to accelerate the Lanczos algorithm? How does the method compare to them?**
>
> - Thanks for the question. We would like to clarify that this task is not the traditional supervised learning task, where we expect the trained network can generalize to unseen test set. The objective here is to obtain the ground state (or wavefunction) of a specific quantum spin system, where the ground state is parameterized by a neural network. For a different system, the ground state is also quite different and it is not expected to optimize a single neural network to represent two different wavefunctions.
> - Regarding the matrix-vector products when calculating the loss (or energy), in practice, we don't need to explicitly construct sparse matrix $H$ and perform matrix-vector products when calculating the local energy $\sum_{j} H_{ij} \frac{\psi\left(c_{j}\right)}{\psi(c_{i})}$, because the typical number of nonzero values in each
> row of $H$ is only of order $N$ ($N$ is the size of spin system). Alternatively, we can use the physical meaning of $H$ and lattice topology to calculate local energy. For example, let's suppose $H = \sum_{<i,j>} \sigma^x_i\sigma^x_j$, where $\sigma^x_i$ is the Pauli matrix applied on spin $i$ and $<i,j>$ denotes the nearest neighbors. Then to calculate local energy for spin configuration $c$, we just need to apply Pauli matrix $\sigma^x_i\sigma^x_j$ on $c$ according to index $i$ and $j$ to get several new spin configurations $c'$. And we evaluate each $\frac{\psi\left(c'\right)}{\psi(c)}$ and sum them together to obtain local energy for $c$. This is how people calculate local energy by using the sparseness of Hamiltonian.
>
> - Lanczos method can only be applied to small systems as it is just a method to efficiently solve eigenvalue problem. Also because of this, for small system, Lanczos method should be way more faster than deep neural ansatz because neural ansatz needs to be trained and use computationally costly MCMC to sample spin configurations during each training iteration. However, for large system, Lanczos method is not even applicable as the size of Hamiltonian matrix growth exponentially. In addition, Lanczos method explicitly solve the ground state vector so that for large system even the ground state itself cannot be stored in any current computer. So in this case, only approximate methods can be used such as variational Monte Carlo with deep neural ansatz.
>
> >**Q5: GNNs can work with arbitrary domains, while here the domains are non-square but are highly structured. So – there is an advantage of GNNs. It is not clear how common are these problems in nature. Is there a lot of interest in such cases? How about unstructured meshes (where, say, the density of the triangles changes across the domain like Finite Element examples)?**
>
> - These four lattices we used are the most commonly studied lattice geometries by the quantum physics community. The ground states of the J1-J2 Heisenberg model are strongly influenced by the underlying geometry of the problem. According to Kochkov et al (2021), all these lattices have complex patterns in the lowest energy spectrum and feature multiple phases, including disordered domains. In addition, triangular and kagome lattices feature geometric frustration and are even harder to model. Moreover, our work has achieved good results on the kagome lattice, where the relative error is at most four times smaller compared to GNN [1]. This is especially noteworthy given that the neural quantum state field has seldom considered the kagome lattice. Understanding the ground state's property of the Heisenberg model on Kagome lattice is one of the main focuses of condensed matter research. However, the ground state of the Kagome Heisenberg model is notoriously difficult to study.
>
> - For more general lattices that can not be directly converted to square lattices will be studied in future work. In these cases, we probably can do convolution in a hierarchy way, which means that we can identify the sublattice first and use our current method on the sublattice to predict a latent representation for each sublattice. And then, we can concatenate the latent embedding of each sublattice to predict the wavefunction value.
>
> [1] Dmitrii Kochkov, Tobias Pfaff, Alvaro Sanchez-Gonzalez, Peter Battaglia, and Bryan K Clark. Learning ground states of quantum hamiltonians with graph networks. arXiv preprint arXiv:2110.06390, 2021.

---

> ### Author Response · Authors · 2023-07-02
> **Response to Reviewer f671 -- Part 4**
>
> >**Q6: It is not clear what is “size” in Table 1. Is it N? So x is in 2^N?**
>
> - Size in Table 1 means how many vertices in the original lattice (or spins in the system). and it is N. Thus, there are $2^N$ different spin configurations in total, and $\boldsymbol{x}$ is one of these $2^N$ configurations.
>
> >**Q7: Based on the citations, it seems that this paper should go to a Physics venue, as the method is quite straightforward in terms of the ML aspects. The method converts a GNN to a CNN-type computation on a non-square but structured grid, and I am sure this was done before in a simpler vision task. Therefore, the methodology is quite straightforward aside from this application, which is relevant mostly to the Physics audience as the citations hint. This is not a major issue, though.**
>
> - Our work is interdisciplinary, focusing on applications of deep learning to quantum physics. Considering AI for science is a quite popular topic these years in ML community, so we believe our work is highly relevant and will be of interest to broader ML community. Compared with previous work, our methodology is simple and effective, which should be the strength of our work. In addition, even though adding virtual vertices appears to be straightforward, it has deeper motivation from this specific quantum problem. First, virtual vertices can gather and distribute the information from the original vertices, which can help increase receptive fields and boost information exchange. Second, we can overlap the same convolution kernel in order to enable information reuse, which is crucial to capture long-range spin correlations [1][2]. We also designed special kernels that can be applied on original lattices and compared with regular kernel on augmented lattices to validate the second point. (More elaboration on this can be found from the third point in answer to Q1 of Reviewer x5EQ)
>
> [1] Y. Levine, O. Sharir, N. Cohen and A. Shashua, Bridging Many-Body Quantum Physics and Deep Learning via Tensor Networks, arXiv: 1803.09780v1.\
> [2] Xiao Liang, Wen-Yuan Liu, Pei-Ze Lin, Guang-Can Guo, Yong-Sheng Zhang, and Lixin He. Solving frustrated quantum many-particle models with convolutional neural networks. Physical Review B, 98(10):104426, 2018.\

---

> ### Author Response · Authors · 2023-07-02
> **Response to Reviewer f671 -- Part 5**
>
> >**Q8: Show the benefits of the approach compared to existing methods. I get and believe that these kinds of CNNs are better than GNNs for the proposed lattices. But - this also should be demonstrated time-wise, as the accuracy difference is quite marginal.**
>
> - Similar to answer in Q2, we would like to clarify that the accuracy difference actually is not marginal. As described in Section 5 "Quantum State Overlap and Energy" part, for the J1-J2 Heisenberg model, approximately improvement on order $1/N^2$ on the energy per spin could be considered significant in approaching the true ground state, where $N$ is the size of the system. And our results can achieve this improvement in most cases. In addition, we present a case study involving a 12 nodes kagome lattice, for which the true ground state can be obtained by exact diagonalization, to illustrate that there could be significant improvements in approaching the true ground state even though the corresponding energy value improvement appears to be marginal. For example, as shown in Figure 4, we observe that even though energy accuracy has a 0.9% improvement (from -0.4492 to -0.4533), which is less than $1/N^2$, the ground state overlap (up to 1) can increase by 10% (from 0.9087 to 0.9971).
>
> - For the time-wise comparison, the answer is similar to Q2. Since GNN doesn't release code, we tried our best to implement GNN to reproduce their results. Computational complexity of variational Monte Carlo is largely determined by the efficiency of evaluating the wave-function amplitudes during MCMC sampling. We test the efficiency on a 36 nodes triangular lattice with a single NVIDIA 2080Ti GPU, and our observation is that for each MCMC step, GNN is ~10 $\times$ slower (GNN is ~4s, ours is ~0.4s) than our method.
>
>
> >**Q9: I would like to see a time comparison with standard methods like Lanczos (+ preconditioning, if possible), implemented on the same GPU (or, let the CNN model run on CPU for a fair comparison).**
>
> - Similar to answer in Q4, Lanczos method can only be applied to small systems as it is just a method to efficiently solve eigenvalue problem. Also because of this, for small system, Lanczos method should be way more faster than deep neural ansatz because neural ansatz needs to be trained and use computationally costly MCMC to sample spin configurations during each training iteration. However, for large system, Lanczos method is not even applicable as the size of Hamiltonian matrix growth exponentially and even the ground state itself cannot be stored in any current computer. So in this case, only approximate methods can be used such as variational Monte Carlo with deep neural ansatz and the ground state is implicitly represented by the neural network.

---

### Review · Reviewer_3pAC · 2023-07-19

**Summary Of Contributions:**

Summary:

They propose lattice convolutions to process non-square lattices by converting them into grid-like augmented lattices through a set of operations. So regular convolution kernel can be applied without using hand-crafted structure encoding, which is needed in the previous GNN method. And this paper design lattice convolutional networks that use self-gating and attention mechanisms to capture channel-wise interdependencies and spatial long-range spin correlations, which contribute to the high expressivity of variational wave functions. This paper
experimentally demonstrate the effectiveness of lattice convolution network wave functions and achieve performance on par or better than existing methods.




**Audience:**

Yes

**Broader Impact Concerns:**

no concerns

**Claims And Evidence:**

No

**Requested Changes:**

Questions:

Q1: From the figure 1, this paper bring some virtual vertex and the virtual lattices gather and distribute the information similar like original vertices. In H1, you only show the ablation study of whether use virtual vertex representation or not. Why in J2 -0.02, the gain is higher than J 0? Additionally, could you provide an ablation study about using virtual lattices or not in feature passing?

Q2:  “Our approach overcomes the shortcomings of previous neural quantum state methods, which not only require extensive prior knowledge but are also designed for a specific lattice or even a specific regime.”  What is the biggest difference or contribution compared with graph neural network? In this approach, for each grid point in different “direction”, the convolution weight is different and learnable. Does it mean LCN is a combination of neighbor grid point?

**Strengths And Weaknesses:**

Strengths:

This paper proposes the first pure deep learning approach that does not require any prior knowledge of quantum physics to solve quantum many-body problems on different types of lattice systems. This paper overcomes the shortcomings of previous neural quantum state methods, which not only require extensive prior knowledge but are also designed for a specific lattice or even a specific regime. This method can be seamlessly applied to different lattices and can still achieve competitive or even better performance than existing methods without introducing prior knowledge. As a result, this method possesses great generalizability in practice, which makes this approach of great value in the study of quantum many-body problems


Weakness:

For a lattice, if it not belongs to square, triangular, Kagome, honeycomb, it is unclear whether this method requires additional cost for incorporating more virtual nodes to allow convolution. Additionally, if a lattice has more vertex, should we increase 3*3 convolution kernel to bigger size?  Thus, Although Lattice Convolution Networks (LCN) do not require hand-crafted design for the model kernel, the lattice pre-processing stage maps the lattice vertices to a template that can be solved by regular convolution kernels, which is also hand-crafted design.

From the experiment, when the Size is bigger, e.g. 100, 108. For the Square Triangular, as the lattice is very regular, the cnn performs better than graph neural network, which seems very make sense for me, but author claim, this kind of ‘regular’ lattice is not the challenge that this paper’s wants to solve.   For the non-square lattice, honeycomb, Kagome lattice, GNN performs better. Thus, the experience is not convincing for non-square challenge solving.

---

> ### Author Response · Authors · 2023-07-27
> **Response to Reviewer 3pAC -- Part 1**
>
> Thank you for your time and valuable comments! We hope your concerns and questions can be addressed by our responses below.
>
> >**Q1: For a lattice, if it not belongs to square, triangular, Kagome, honeycomb, it is unclear whether this method requires additional cost for incorporating more virtual nodes to allow convolution. Additionally, if a lattice has more vertex, should we increase 3*3 convolution kernel to bigger size? Thus, Although Lattice Convolution Networks (LCN) do not require hand-crafted design for the model kernel, the lattice pre-processing stage maps the lattice vertices to a template that can be solved by regular convolution kernels, which is also hand-crafted design.**
>
> - Thank you for the question. Similar to the answer of Q5 to reviewer f671, we would like to first emphasize the importance of these four lattices we studied. These four lattices we used are the most commonly studied lattice geometries by the quantum physics community. The ground states of the J1-J2 Heisenberg model are strongly influenced by the underlying geometry of the problem. According to Kochkov et al (2021), all these four lattices have complex patterns in the lowest energy spectrum and feature multiple phases, including disordered domains. In addition, triangular and kagome lattices feature geometric frustration and are even harder to model. Thus, developing a general method that can learn ground states for these four kinds of most important lattces is already an important contribution, considering that most previous methods only focus on one kind of lattice.
>
> - For more general lattices that can not be directly converted to square lattices, they are beyond the scope of this work and will be considered in future work. In these cases, we probably can do convolution in a hierarchy way, which means that we can identify the sublattice first and use our current method on the sublattice to predict a latent representation for each sublattice. And then, we can concatenate the latent embedding of each sublattice to predict the wavefunction value.
>
> - After converting lattices to grid-like structure, we can seamlessly use any CNN techniques. We did try to increase the kernel size to $5 \times 5$ for larger lattices in early stage experiments, but our observation was that the performance is about the same with $3 \times 3$. The reason could be that we use non-local blocks that can already effectively capture long-range interaction, so we don't need to increase kernel size to enlarge receptive field.
>
> - Lattice augmentation is a preprocessing step but is simple and general for these four kinds of lattices, which does not need any prior physical knowledge. What we refer to about hand-crafted design is the complicated sublattice encoding in GNN [1], which needs to leverage physical knowledge to design different sublattice encoding for different kinds of lattices and even different J2 values in the Hamiltonian.
>
> [1] Dmitrii Kochkov, Tobias Pfaff, Alvaro Sanchez-Gonzalez, Peter Battaglia, and Bryan K Clark. Learning ground states of quantum hamiltonians with graph networks. arXiv preprint arXiv:2110.06390, 2021.
>
> >**Q2: From the experiment, when the Size is bigger, e.g. 100, 108. For the Square Triangular, as the lattice is very regular, the cnn performs better than graph neural network, which seems very make sense for me, but author claim, this kind of ‘regular’ lattice is not the challenge that this paper’s wants to solve. For the non-square lattice, honeycomb, Kagome lattice, GNN performs better. Thus, the experience is not convincing for non-square challenge solving.**
>
> - Thanks for the question. Our approach consistently delivers performance either on par with, or superior to, GNN when applied to a honeycomb lattice of sizes 32 and 98. Despite that ground state of the Kagome Heisenberg model is notoriously difficult to study in quantum, our method still outperform GNN on kagome lattice of size 36 and is competitive with GNN on kagome lattice of size 108. Hence, our claim of comparable or superior performance to GNNs is substantiated. Furthermore, considering our method eliminates the need for prior knowledge in designing hand-crafted sublattice encoding, it offers increased practical utility. This is especially beneficial when prior knowledge is lacking, thereby enhancing the versatility and applicability of our approach.

---

> ### Author Response · Authors · 2023-07-27
> **Response to Reviewer 3pAC -- Part 2**
>
> >**Q3:  From the figure 1, this paper bring some virtual vertex and the virtual lattices gather and distribute the information similar like original vertices. In H1, you only show the ablation study of whether use virtual vertex representation or not. Why in J2 -0.02, the gain is higher than J 0? Additionally, could you provide an ablation study about using virtual lattices or not in feature passing?**
>
> - Thank you for the question. We would like to clarify that Appendix H1 is not about ablation study of whether use virtual vertex representation or not. Appendix H1 shows the ablation study of whether using mask operation to reset the vertices outside the original lattice area.
>
> - J2=-0.02 means interaction between next nearest neighbors are involved in the Hamiltonian so that the interactions are more intricate and the ground state is harder to model. About the reason why performance drops more without masking operation in the J2=-0.02 case, we hypothesize that masking operation can force the model focus only on the original lattice area, and eliminate the effect of vertices that shouldn't contribute to the ground state modeling. In this way, the model can learn the ground state more effectively in these harder cases.
>
> - As suggested by the reviewer, we perform an ablation study about whether to use virtual lattices or not in feature passing and added the result in Appendix H2. Specifically, in our original design, virtual vertices participate in the convolution in the same way as the original vertices. As an ablation study, we reset virtual vertices value to zero after each convolution so that virtual vertices do not participate in the feature passing. The result is shown in the table below. We can see that using virtual vertices in feature passing can improve the performance.
>
> | Lattice | Size | J2 | Virtual vertices in feature passing or not  | Energy per spin |
> | ----- | ----- | ----- | ----- | ----- |
> | `Kagome` | 36 | 0 | No  | -0.4321(5) |
> | `Kagome` | 36 | 0 | Yes | **-0.4367(4)** |
> | `Kagome` | 36 | -0.02 | No | -0.4354(5) |
> | `Kagome` | 36 | -0.02 | Yes | **-0.4384(5)** |
>
> >**Q4: “Our approach overcomes the shortcomings of previous neural quantum state methods, which not only require extensive prior knowledge but are also designed for a specific lattice or even a specific regime.” What is the biggest difference or contribution compared with graph neural network? In this approach, for each grid point in different “direction”, the convolution weight is different and learnable. Does it mean LCN is a combination of neighbor grid point?**
>
> - GNN features isotropic weights, which prevent it from capturing rich structure information, thereby it needs to design sublattice encoding to augment the original spin configuration input on lattices. However, sublattice encodings are not easy and straightforward to develop, it requires complated prior physical knowledge and need to be tailored to different lattices or even different regime (J2 value) for the same lattice. Distinct from GNN, we make the key observation that these non-square lattices studied by GNN feature repetitive local patterns and can be converted to grid-like lattices in a simple and principled way. Thus, CNN can be applied without using any sublattice encoding. By using CNN, complex patterns and interactions are automatically learnt in the kernel space.

---

### Author Response · Authors · 2023-07-27
**General response**

We would like to thank the reviewers for their time and valuable comments on our work. We revised the manuscript with addtional experiments as suggested by reviewers and provided point-to-point responses to questions of reviewers in separate replies. Please let us know if there is any further concern or questions.

We summarize the main modifications of the manuscript below.

- Added citation of 3D shape analysis paper mentioned by reviewer x5EQ in the "Relationship with Prior Work" part (in Section 1) and highlight the difference with our work.
- Added citation of lattice gauge equivariant paper mentioned by reviewer x5EQ in Section 2.3 (Related Work).
- Clarified the definition of $\psi(c)$ Equation (1), as suggested by reviewer x5EQ.
- Modified the caption of Fig 3 to make it more clear, as suggested by reviewer f671.
- Added ablation study about whether to use virtual vertices in feature passing in Appendix H.2, as suggested by reviewer 3pAC.

---

### Decision · Action_Editors · 2023-08-27

**Recommendation:** Reject

**Comment:**

Thank you for submitting your manuscript to TMLR. We have received reviews from three experts in the field, and after careful consideration, we have decided to request a resubmission of major revision to your manuscript before considering it for publication.

The reviewers have raised several concerns, and we would like you to address these points in your revision. For example, while there might be good contribution to the specific domain (e.g., Learning Ground States of Quantum Many-Body Systems), the general technical implication of your paper to the machine learning field is unclear. For another example, the quantitative comparisons with some highly relevant baselines are missing.  You have primarily argued that DMRG or MPS has an established track record of success, particularly within specific problem classes like one-dimensional quantum systems with limited entanglement but has limitations in capturing high entanglement in 2D systems. However, there is no quantitative evidence showcasing how the proposed method surpasses DMRG and MPS in terms of both accuracy and runtime efficiency. In addition, there are quite a few other critical comments that need to be addressed. Please provide a detailed explanation on your revision and how it addresses these comments.

**Audience:**

This paper is relevant to TMLR, but might not be very popular because it is related to a very specific application domain - Learning Ground States of Quantum Many-Body Systems

**Claims And Evidence:**

The authors have proposed a new method for learning ground states of quantum many-body systems, and experimentally evaluated its performance. However, the reviewers have raised some concerns about the experimental setting, and the lack of quantitative comparison with some highly relevant baselines.

**Resubmission Of Major Revision:**

The authors may consider submitting a major revision at a later time.